# TOOLFUZZ: AUTOMATED AGENT TOOL TESTING

## ABSTRACT

Large Language Model Agents (LLM Agents) leverage the advanced reasoning capabilities of LLMs in real-world applications. To interact with the environment, these agents require tools such as web searches or database APIs. As the agent provides the LLM with tools documentation alongside a user query, the completeness and correctness of this documentation is critical. However, tool documentation is often over-, under-, or ill-specified, impeding the agent's accuracy. Standard software testing approaches struggle to identify these errors as the documentation is usually in natural language. Thus, despite its importance, there currently exists no automated method to test agent tool usage. To address this, we present TOOLFUZZ, the first automated agent tool testing method. TOOLFUZZ combines LLMs with fuzzing to generate diverse and natural user queries causing tool runtime errors or semantically incorrect agent responses. We evaluate TOOLFUZZ on 139 tools from the community based open source LangChain and the production ready closed source Composio and find that all LangChain tools and the majority of Composio tools are erroneous. To validate the relevance of errors, identified by TOOLFUZZ, we design an automated pipeline to improve tool documentation. Specifically, we introduce two novel benchmarks—over 300 tasks, known ground truth, and real environments based on GitHub and terminal file management. Our automated tool-fixing pipeline increases accuracy from 22.9% to 35.4% on GitHub tasks and from 29% to 39% on file management tasks. TOOLFUZZ consistently outperforms the baselines and identifies 50% more unique errors while reducing the False Discovery Rate by $4.5\times$, making it a key component for building reliable AI agents.

## 1 INTRODUCTION

LLM agents and compound systems (Yao et al., 2023b; Zaharia et al., 2024) aim to combine the powerful reasoning capabilities of LLMs with real-world interactions to solve complex tasks. An agent receives a natural language query from a user and performs a sequence of actions that interact with external *tools* performing web search Zhou et al. (2024), code execution (Yang et al., 2024c), or querying a database (Wang et al., 2024), with the goal of solving the specified task. As such, effective interaction with these tools is critical for successful task completion.

**Reliability Problems of Agent Tools**    To facilitate tool usage for agents, each tool is accompanied by documentation detailing its functionality and intended use, which can be provided to the agent LLM as part of the prompt. In practice, however, LLM agents often fail to use the available tools correctly (Balunovic, 2024; Sun et al., 2024; Yang et al., 2024b) because their documentation assumes human-level understanding. Thus, it is often underspecified (leaving crucial details implicit), overspecified (focusing narrowly on a single use case despite broader applicability), or illspecified (where the tool's functionality is not aligned with its documentation). Although tools play a crucial role in agent systems (Qu et al., 2024; Yuan et al., 2024), automated methods for detecting such errors are lacking. Traditional fuzzing techniques, generating seemingly meaningless random strings, fall short for agent testing, as they cannot simulate the expected natural language queries. For instance, when using the well-known fuzzing algorithm American Fuzzy Lop (Zalewski) on an agent tool for several hours, it failed to find even a single example that realistically represents an LLM agent scenario.

**Testing Agent Tools with TOOLFUZZ**    To address this, we introduce TOOLFUZZ, a novel method for automatic end-to-end tool testing. TOOLFUZZ employs two techniques to uncover specification

Figure 1: Overview of the two error detection techniques of TOOLFUZZ, consisting of (1) a fuzzing based approach and (2) an invariance based approach utilizing consistency checks. Prompts are denoted by $p$ or $p_j$, tool calls by $I_p$ or $I_j$, tool responses by $O_p$ or $O_j$ and agent responses by $a$ or $a_j$.

errors: (1) by generating queries that lead to tool runtime errors, achieved by combining fuzzing techniques with LLM-based query generation, and (2) by generating queries that result in incorrect agent responses, using synonymous prompt generation and a series of cascading consistency and correctness checks at various stages of the agent's processing. Experimentally, we show that TOOLFUZZ successfully identifies a large number of erroneous queries, aiding in the improvement of the tool and therefore its utility.

**Main Contributions:**

- A novel, end-to-end agent-centric method for finding errors in tools, called TOOLFUZZ.

- A new benchmark suite that focuses on evaluating accurate tool utilization for file management and GitHub agents, emphasizing precise tool invocation rather than sophisticated reasoning and planning.

- A thorough experimental evaluation of TOOLFUZZ across a wide variety of community agent tools from LangChain (Chase, 2022) and production ready agent tools from Composio (Composio, 2025).

## 2 BACKGROUND AND RELATED WORK

In this section, we review the most relevant work.

**Language Models** Throughout this work, we rely on Large Language Models (LLMs). Specifically, we use GPT-4 (OpenAI et al., 2024), GPT-4o, GPT-4o-mini (OpenAI, 2024), Claude 3.5 Haiku (Anthropic, 2024) and Gemini 2.0 Flash (DeepMind, 2025) which have shown impressive performance across various challenging benchmarks. Despite this, they remain susceptible to "hallucinations" that can undermine their trustworthiness (Lee et al., 2023; Manakul et al., 2023; Mündler et al., 2024). To address this issue, numerous methods have emerged, employing various prompting techniques (Mündler et al., 2024; Fluri et al., 2023; Wang et al., 2023b). Importantly, underspecified documentation promotes hallucinations which may lead to false positives in our setting. Additionally, LLMs are used as correctess oracles (Wei et al., 2023; Zheng et al., 2023) to reduce hallucinations. Furthermore, cross-checking multiple generations with assertive generations (Chen et al., 2022) can help balance efficiency with more robust correctness verification.

**LLM Agents** LLM agents enable LLMs to interact with external tools such as Web-APIs, databases, and code execution environments. The LLM is responsible for reasoning, planning, and tool usage to continuously enrich its context, thereby improving the quality of the final response (Shinn et al., 2023; Wang et al., 2023a; Yao et al., 2023b). An agent system $\mathcal{A}$ consists of a language model $L$ which has access to a set of tools $\mathcal{F} \ni f_i$. As many tools are frequently wrappers around robust Web-APIs or other well-tested libraries, we focus mainly, but not only, on potential failures due to

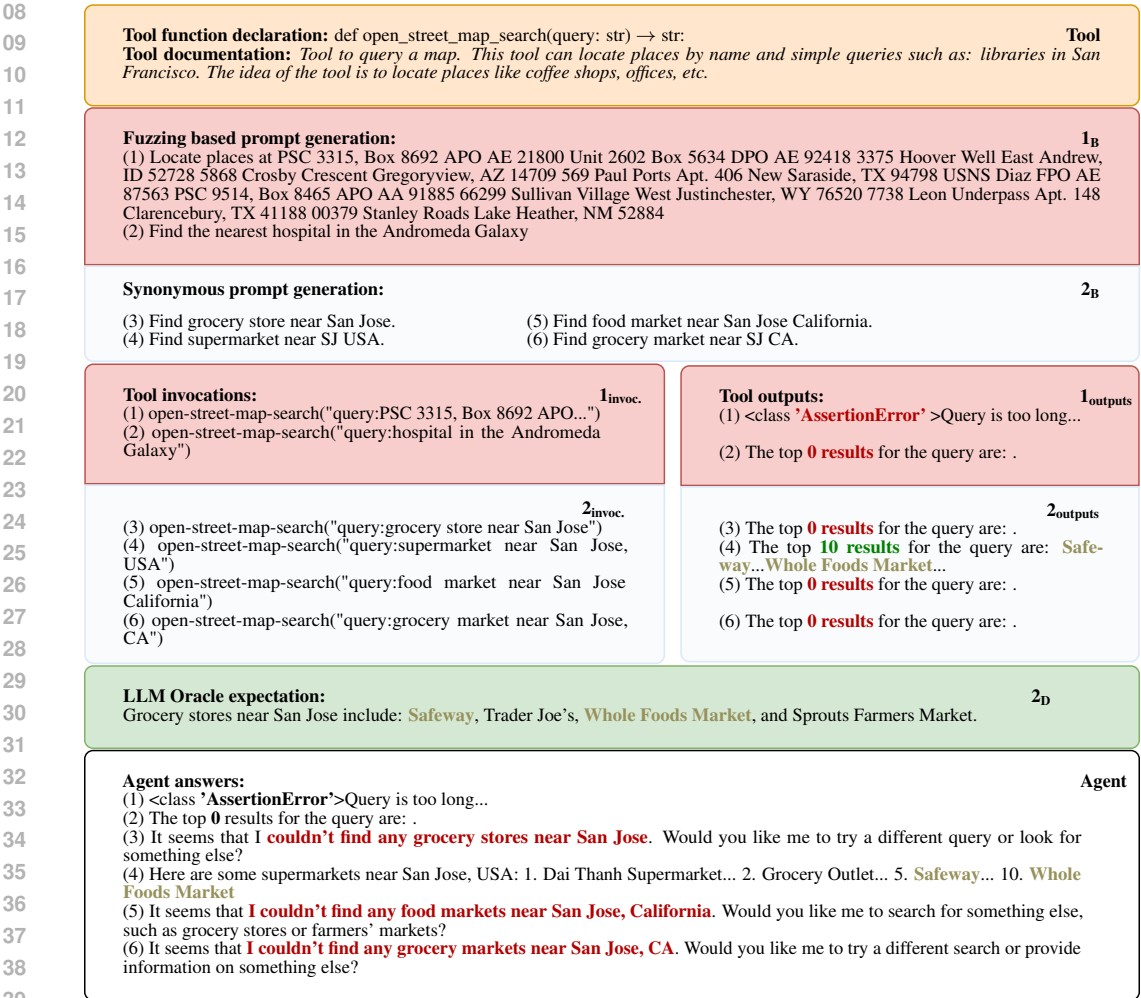

Figure 2: Input/Output overview for open_street_map_search tool evaluated with TOOLFUZZ. Note that the numbering corresponds to the numbering of the two approaches in Figure 1.

incomplete or erroneous documentation that would undermine the tools utility to the agent (Qu et al., 2024; Yuan et al., 2024).

**Agent Benchmarks and Tool Usage**   Various benchmarks have been developed to assess the capabilities of LLM agents across diverse tasks such as long-horizon planning (Liu et al., 2023), security analysis (Debenedetti et al., 2024; Naihin et al., 2023; Ruan et al., 2024), real-world web interactions (Deng et al., 2023a; Zhou et al., 2024; Yao et al., 2023a) and multimodal agents (Xie et al., 2024). Further, there is a series of benchmarks that focus on function call accuracy (F. Yan et al.; Huang et al., 2024b; Tang et al., 2023; Xu et al., 2023). However, they primarily focus on the reasoning and planning capabilities of the LLM and not on the correctness of the tools and the tool-agent interaction. While methods for enhancing tool documentation (Qu et al., 2024; Yuan et al., 2024) recognize tools and their documentation as critical components, they do not provide a methodology to identify such insufficiencies. Our work addresses this gap by introducing TOOLFUZZ, along with two custom benchmarks, a GitHub and a File Management benchmark, that specifically focus on evaluating end-to-end agent-tool interactions over the planning capabilities of LLMs.

**Fuzz Testing**   In classical software engineering, fuzzing is a standard technique to discover errors or vulnerabilities in software by testing on random or semi-random inputs (Wen, 2024; Zeller et al., 2024). White-box fuzzing leverages source code insights, black-box fuzzing treats the program as a black box, and gray-box testing blends both approaches. Recently, LLMs have been employed to enhance fuzzing in creating more semantically meaningful tests (Deng et al., 2023b; Huang

et al., 2024a; Yang et al., 2024a). However, both traditional and modern methods must be carefully managed, as a high number of false positives undermines the utility of the testing process. While fully LLM-driven fuzzing faces challenges in cost, scope, and reliability (Jiang et al., 2024), combining fuzzing with machine learning has proven effective for testing (He et al., 2019). Consequently, we adopt a hybrid approach: instead of producing purely random strings, which would likely be out of distribution for the LLM driven agent, we combine fuzzing and LLM-based prompt generation, using partial knowledge of the tool's semantics to systematically investigate the agent-tool interactions.

## 3 OVERVIEW

We now provide an overview of TOOLFUZZ. First, we explain the types of documentation errors: under-, over-, and ill-specification as well as the different types of agent failures, specifically tool runtime and correctness errors. Then, we describe our techniques for finding user prompts, tailored to each type of agent failure.

**Documentation Errors** Several errors can arise due to faulty documentation. Recent work has shown that in practice, agents often call tools inaccurately (F. Yan et al.; ScaleAI; Yang et al., 2024c; Yao et al., 2024). As these tool calls primarily depend on the tools' documentation, it can be held accountable for a large portion of agent failures. First, documentation can be *underspecified* regarding a tool's limitations, causing the LLM to use the tool in unsupported contexts, such as using arXiv to search for "Italian food". Additionally, underspecification can restrict tool usage, i.e. the pubmed tool has additional features beyond searching for paper titles which are unspecified in the documentation (see more in §5.5). Second, documentation can be *overspecified*, artificially reducing the scope or the ways in which the tool can be used. Third, documentation can be *illspecified*, reflecting a misalignment between a tool's functionality and its described usage. In practice, under- or ill-specified argument documentation can include incomplete or incorrect enumeration values, missing or outdated syntax details, or omitted relational constraints. These documentation errors often lead to tool misuse, highlighting the need for methods to detect these shortcomings.

**Agent Tool Failures** Initial investigation reveals two distinct manifestations of these documentation errors in agent tool failures: first, runtime failures, where invalid or misformatted inputs lead to runtime tool errors, and second, correctness errors, where the tool returns incorrect results for a given user query. These issues often arise from a mismatch between tool documentation, written from the perspective of developers with their own context and how it is interpreted by LLMs.

### 3.1 TOOLFUZZ

To address the two types of agent tool failures TOOLFUZZ utilizes two custom techniques to detect errors: (1) a fuzzing inspired approach to detect runtime errors, and (2) an invariance-based approach to detect correctness errors. Both techniques of TOOLFUZZ are illustrated in Figure 1. In the following, we provide a detailed overview of these techniques using the open-street-map-search tool, specified in Figure 2, as a running example. The tool processes free-form text queries to retrieve location information.

**Runtime Failure Detection** The runtime failure detection technique combines fuzzing with LLM-based generation to produce realistic testing prompts (see Figure 1.1). For fuzzing, we use our TaintFuzzer (Figure $1.1_A$) to stress test the tool, collecting tool inputs that lead to runtime errors. In the case of the open-street-map-search tool, which has a length restriction on the query parameter, the fuzzer identifies a particularly long query, *"PSC 3315, Box 8692 APO AE 21800 Unit 2602 Box 5634 DPO AE 92418..."*, causing a runtime error. This failing input is then passed to an LLM along with the tool documentation to generate a user query incorporating it. For our running example, the generated user query is *"(5) Locate places at PSC..."* (Figure $2.1_B$).

The generated query is then passed to the agent's LLM (Figure $1.1_B$), which plans a tool invocation as shown in Figure $2.1_{invoc.}$. The tool execution is then monitored for runtime errors. In our example, the tool raises *"(5) <class 'Assertion Error'>Query is too long..."* (Figure $2.1_{outputs}$). Whenever such an error occurs, the corresponding query is collected as an erroneous prompt (Figure $1.1_D$).

```
@tool('open-street-map-search')
def open_street_map_search(query: str) -> str:
    assert len(query) < 100, 'Query is too long.
    Query must be less than 100 characters'

    search_query = query.split('query: ')[1]
    keywords = ['supermarket', 'coffee shop', 'library', 'office']

    if any(word in search_query for word in keywords):
        response = request(f"{OPEN_STREET_BUILDING_SEARCH}{query}")
    else:
        response = request(f"{OPEN_STREET_NATURE_SEARCH}{query}")
    response_json = response.json()
    return response_json
```

**Documentation:**
A tool to query a map, capable of
locating places by name and handling
simple queries. The queries must start
with 'query: ', an example is 'query:
libraries in San Francisco.' The
purpose of the tool is to locate places
like coffee shops, offices, etc.

(b) Documentation

(a) Code

Figure 3: Example Implementation of the `open-street-map-search` tool.

**Correctness Failure Detection**  To detect correctness failures, TOOLFUZZ generates synonymous prompts and applies cascading checks throughout the agent process, including checks for tool argument consistency, tool output consistency, and an LLM correctness evaluation (Figure 1.2). First, TOOLFUZZ generates a template prompt, which is then populated with synonymous mask infills (Figure 1.2 A and B). For example, "Find [A] near [B]." can be instantiated with strings like "Find grocery store near San Jose." and "Find supermarket near SJ USA" (Figure 2.2$_{\text{B}}$).

Once these $n$ user queries $p_1, \ldots, p_n$ are generated, they are passed to the agent LLM (Figure 1.2$_{\text{B}}$), which again plans and executes tool invocations. Here, TOOLFUZZ checks the inputs $I_1, \ldots, I_n$ and outputs $O_1, \ldots, O_n$ separately for consistency (Figure 1.2$_{\text{C}}$). For our running example, these consistency checks fail as not only the tool inputs $I_j$ generated by the agent LLM are not equal, but the tool outputs $O_j$ also do not match each other. Specifically, in Figure 2.2$_{\text{invoc.}}$, we see that the input arguments do not coincide. Similarly, in Figure 2.2$_{\text{outputs}}$, we see that the responses are also not equal.

However, such checks alone may miss consistent outputs, which are obviously wrong. To catch these, TOOLFUZZ uses an LLM correctness evaluator (an LLM Oracle) to compare the agent's output against a generated expectation (Figure 1.2$_{\text{D}}$). In our example, the LLM Oracle expects *"Grocery stores near San Jose include: Safeway..."*, but the agent instead replies *"It seems that I couldn't find any grocery stores near San Jose..."* (Figure 2.Agent). Since this response contradicts the expected answer (Figure 2.2$_{\text{D}}$), TOOLFUZZ classifies it as incorrect.

TOOLFUZZ utilizes all three checks to reduce the number of false positives. The first two metrics expose description insufficiencies, while the LLM Oracle reduces the set of positives further, to a set of positives with mismatched expectations. The resulting set will be added to the erroneous prompts.

## 4 METHODOLOGY

As discussed in §3, we identify two main types of tool failures: *runtime tool failures* and *correctness failures*. We now explain the two techniques to detect these in detail.

### 4.1 RUNTIME TOOL FAILURES

Runtime tool failures of the agent occur when inputs break the tool under test, $f_{\text{tested}}$. To generate such inputs, we employ a two-stage process: first, a Taint Fuzzer generates inputs to trigger runtime errors for the tool in isolation (Figure 1.1). Then, provided that the inputs pass a sanity check against the documentation, an LLM is used to create natural user queries for evaluation on $\mathcal{A}$ (Figure 1.1$_{\text{B}}$). As the Taint Fuzzer operates independently of the tool documentation, this second step is crucial to minimize false positives.

**Taint Fuzzing**  As most tools have strong syntax or semantic priors on their input, it is essential to capture those and ensure that the fuzzer samples accordingly. An example of a syntax prior is given in Figure 3a, where the input `query` is required to start with the text "query: ". A semantic prior could be formatted like an address. To generate inputs satisfying a specific syntax, we analyze the tool using

taint object analysis and collect syntax requirements like specific JSON or CSV formats, or string splitting operations. This is then used by our custom generator, which generates syntax-conforming arguments based on their types and syntactic patterns. To generate inputs with a semantic prior, we leverage resources such as dictionaries or LMs. If the generated arguments lead to runtime tool errors, they are collected for future prompt generation. For the `open-street-map-search` tool, the fuzzer collects all generated arguments exceeding 100 characters (Figure 3a).

**Prompt Generation**   While the fuzzer generates valid words, numbers, etc., these arguments may still not satisfy the requirements specified in the tool documentation.   For example, `open-street-map-search` is designed and specified for finding locations, not scientific papers. To mitigate this issue, we conduct a sanity check against the documentation by instructing the agent's LLM to invoke the tool with the previously found arguments, given the documentation. Refusal of the LLM indicates that the arguments do not satisfy the documented requirements. If the sanity check passes, a natural language prompt for $\mathcal{A}$ is generated via an LLM, based on the tool documentation and the arguments that break the tool. Finally, we filter for the user queries that lead to runtime tool errors when passed to the agent $\mathcal{A}$.

### 4.2   Correctness failure detection

Unlike runtime failures, the primary challenge in detecting correctness failures is assessing the outputs of $f_{\text{tested}}$ in the absence of ground-truth data. We address this by introducing prompt sets $\mathbb{P} = \{p_1, p_2, \ldots, p_n\}$, consisting of synonymous prompts. The idea is that synonymous user queries will lead to synonymous agent responses given the correct tool documentation. Thus, a violation of this indicates faulty tool documentation. To check this, we employ a cascade of checks: we check whether synonymous user queries $p_1, \ldots, p_n$ are (i) mapped to equivalent tool calls $I_1, \ldots, I_n$, and (ii) equivalent tool outputs $O_1, \ldots, O_n$ and (iii) check if the output appears plausible to an LLM. We find that if all these checks fail, we have sufficient evidence to treat this prompt as erroneous.

**Prompt Generation**   Generating the synonymous prompt set $\mathbb{P}$ involves multiple steps. First, an LLM ($LM_{\text{prgen}}$) creates a template question using the tool documentation, e.g., *"Find [venue] in [city]"* (Figure 3a). Utilizing the tool documentation, an LLM then generates synonymous infills for the masked words, forming $\mathbb{P}$. For our example, infills for *"[venue]"* can include [*'libraries', 'public libraries', 'city libraries'*], while infills for the mask *"[city]"* can be [*"SF", "San Francisco", "San Francisco CA"*]. Multiple prompts and infills are generated per tool.

**Correctness Detection**   Next, ToolFuzz invokes the agent $\mathcal{A}$ on $\mathbb{P}$ and collects the tool inputs $I_j$ and outputs $O_j$. The key property for correctness detection is that synonymous prompts should result in consistent agent responses given the correct tool documentation. Thus, inconsistent tool invocations are likely to lead to inconsistent tool responses leading to erroneous agent responses.

- **Input Evaluation** Specifically, we check the *input consistency* of a prompt set $\mathbb{P}$ by verifying if the values of the argument across all function inputs $I_1, \ldots, I_n$ are equal to ensure identical function calls. An input consistency check failure indicates underspecification of the tool arguments.

- **Output Evaluation** Analogously, we check the *output consistency* of $\mathbb{P}$ by comparing the responses $O_1, \ldots, O_n$, again via exact matching. The output consistency serves as a proxy for comparing the inherently challenging natural language agent responses.

**LLM Oracle**   In practice, a lot of tools are non-deterministic, i.e., news APIs or search engines, where the same input can lead to different outputs.  To combat this, we add a third check for plausibility. While each consistency check on its own is insufficient, when combined they reduce the FDR to an acceptable level: In the absence of ground truth, we ask an LLM to answer the queries $\mathbb{P}$ (Mündler et al., 2024), followed by majority voting. The majority answer is then compared to the agent responses by an LLM Oracle, rating their similarity on a scale from 1 to 10, with 5 as the threshold (Zheng et al., 2023). We note that, even though some tools require private API access, the expected answer from the majority vote is most often sufficient. We consider $\{p_j\}_j$ to be faulty if all checks fail simultaneously.

## 5 EXPERIMENTAL EVALUATION

In this section, we demonstrate TOOLFUZZ's effectiveness at error detection. We evaluate TOOLFUZZ on a wide set of tools and agents and show that it outperforms baselines based on prompt engineering. Additionally, we introduce two new agent benchmarks alongside an automated pipeline for improving tool documentation to show the utility of the erroneous prompts identified by TOOLFUZZ. Finally, we demonstrate TOOLFUZZ's utility in combination with existing tool description fixing approaches.

### 5.1 EXPERIMENTAL SETUP

We now describe the experimental setup we used, including the agent, tools, metrics, and baselines.

**Agent** Our method is applicable to any agentic paradigm and it consistently discovers errors in the tools under test. We mainly evaluate using ReAct (Yao et al., 2023b), but also evaluate on OpenAI functions and tool calling (Chase, 2022). As a LLM, we mainly use GPT-4o (OpenAI et al., 2024). Evaluation with additional models like GPT-4o-mini (OpenAI, 2024), Claude 3.5 Haiku (Anthropic, 2024) and Gemini 2.0 Flash (DeepMind, 2025) can be found in App. H.1.

**Tools** We selected the 56 LangChain Community tools from a total of 96 tools which do not require API keys to facilitate reproducability and adoption. Additionally, we have selected 83 tools from production ready closed source Composio (Composio, 2025) library, where we again selected tools that do not require API keys. More details are available in App. E.

**Metrics** To assess TOOLFUZZ's effectiveness, in finding errors, our main metric is the number of unique errors found. We report unique runtime errors, identified by grouping the same errors at runtime, and unique correctness errors, which require manual grouping. Further, to facilitate comparison, we report the number of unique errors (UE) that are found within a given time budget of 5 minutes and the number of tool calls per unique error (TC/UE). The number of unique errors for a given token budget can be found in App. D. As the TC/UE for correctness errors close to $n$, where $n$ is the number of synonymous prompts we report instead the very important false dicovery rate FDR $= \frac{\text{FP}}{\text{FP+TP}}$. Finally, we assess the power of a method by the number of total unique errors this method can find, estimated by the standard Chao1 estimator (Chao, 1984; Colwell, 2009) and report its 95% confidence interval (TUE-CI).

**Prompt Engineering Approaches** Without prior research on testing agent tools, we introduce two prompt engineering approaches based on GPT-4o OpenAI et al. (2024), each assessing tool runtime and correctness errors. The first baseline relies on *gray-box testing* (BG), where an LLM generates test prompts solely from the tool's name and documentation. We create two prompt variants, one for runtime failure detection and another for correctness failure detection App. J.4 to generate user queries which are then tested. Additionally, for correctness failure detection we use an LLM judge to flag incorrect agent's response. The second method relies on *white-box testing* (BW) mirroring the gray-box approach but additionally incorporates the tool's source code during prompt generation.

### 5.2 EVALUATION ON TOOLS

We first evaluate of TOOLFUZZ's capabilities to detect runtime and correctness errors in the tools and present the results in Table 1. We evaluated on both, LangChain community tools (LC) and Composio tools (C). The LLM used here is GPT-4o with temperature set to 0 and the ReAct agent paradigm.

**Evaluating Runtime Failure Detection** We see that TOOLFUZZ outperforms the graybox (BG) and whitebox (BW) baseline methods in all metrics. Further, we ablation for the sanity check shows not only an improvement in the number of unique errors found, but also a reduction in the number of tool calls per unique error (TC/UE). Finally, we see that TOOLFUZZ is more powerful compared to the other methods, as the lower bound of the total number of unique errors (TUE-CI) is significantly higher than the upper bounds on LangChain, and still a lot higher than the lower bounds on Composio. As Composio tools are commercial and production ready, they are on average of higher quality, which is reflected in the lower number of unique errors found per tool tested (56 for LangChain, 83 for Composio). We note that the lower bound of the TUE-CI is higher than the number of unique errors,

Table 1: Comparison of TOOLFUZZ (TF) to Baseline Whitebox (BW) and Baseline Graybox (BG) in both Runtime error detection and Correctness error detection. LC - Langchain tools, C - Composio, UE - Number of Unique errors, TC/UE - tool calls per unique error, TUE-CI - 95% confidence interval for the total number of unique errors (Chao1 with bias correction), False Positive Rate - FDR.

| | Runtime error detection (Figure 1.1) | | | | | | Correctness errors detection (Figure 1.2) | | | | | |
| --- | --- | --- | --- | --- | --- | --- | --- | --- | --- | --- | --- | --- |
| | UE ↑ | | TC/UE ↓ | | TUE-CI ↑ | | UE ↑ | | FDR ↓ | | TUE-CI ↑ | |
| | LC | C | LC | C | LC | C | LC | C | LC | C | LC | C |
| BW | 35 | 63 | 496 | 440 | [35, 42] | [73, 117] | 14 | 21 | 0.68 | 0.81 | [24, 47] | [41, 87] |
| BG | 36 | 57 | 105 | 128 | [36, 38] | [57, 63] | 30 | 32 | 0.85 | 0.79 | [16, 30] | [29, 63] |
| TF$_{SC}$ | 57 | 55 | 92 | 106 | [59, 70] | [68, 106] | - | - | - | - | - | - |
| TF$_{CC}$ | - | - | - | - | - | - | 50 | 54 | 0.43 | 0.48 | [59, 109] | [63, 86] |
| TF$_{LLM}$ | - | - | - | - | - | - | 48 | **54** | 0.36 | 0.47 | [**67**, 143] | [**64**, 90] |
| TF | **64** | **78** | **88** | **101** | [**76**, 124] | [**85**, 108] | 42 | 45 | **0.15** | **0.37** | [57, 124] | [58, 92] |

as it is a lower bound on the total number of unique errors discoverable by TOOLFUZZ and thus neccessarily higher than the number of unique errors found in the time budget.

**Evaluating Correctness Detection**   We see that all variants of TOOLFUZZ outperform the baseline methods significantly in all metrics. As a low False Discovery Rate (FDR) is crucial for the utility of a fuzzer, it is crucial to minimize the FDR as previously discussed in §4.2. We see that TOOLFUZZ achives a much lower FDR compared to its variants TOOLFUZZ-CC, which uses only consistency checks and TOOLFUZZ-LLM, which uses only the LLM oracle. While the combination of the two methods neccessarily reduces the number of unique errors (UE), it reduces significantly the FDR, which is in practice a favorable trade-off.

## 5.3 UTILITY OF TOOLFUZZ FOR IMPROVING TOOL DOCUMENTATION

**Benchmarks**   In order to evaluated TOOLFUZZ utility for improving tool documentation, we evaluate the performance of the agent in two custom benchmarks as existing benchmarks focus primarily on LLM reasoning and planning and overlook the importance of tools. We created two two new

Table 2: Comparing the accuracy (Pass Rate) for TOOLFUZZ Auto Fix (TF-AF), DRAFT and DRAFT applied on top of TOOLFUZZ Auto Fix for the File Management (FMB) and GitHub Benchmarks (GHB)

| | Tool | Original | TF-AF | DRAFT | TF-AF + DRAFT |
| --- | --- | --- | --- | --- | --- |
| FMB | Terminal | 0.29 | **0.39** | 0.38 | 0.32 |
| | FileToolkit | 0.33 | **0.4** | 0.35 | 0.34 |
| GHB | GithubToolkit | 0.22 | **0.35** | 0.29 | 0.25 |

benchmarks for evaluating the performance of agents in end-to-end real world settings. The first benchmark, called *File Management Benchmark* (FMB), includes 42 folder structures and 257 tasks, accompanied by scripts for setup and validation. The second, *GitHub Benchmark* (GHB), features 54 tasks within a single repository, divided into six categories, with built-in logic for task validation and environment reset. Both benchmarks are divided into validation and test sets to avoid selection bias.

**Automatic Documentation Fixing**   We evaluate the utility of TOOLFUZZ for our autofix pipeline TOOLFUZZ-AutoFix (TF-AF). TF-AF uses erroneous prompts identified by TOOLFUZZ to improve tool documentation, via argumentative interactive prompting (de Wynter & Yuan, 2024). We select the best documentation fix out of 10 fixes on the validation set, and report performance on the test set.

**Evaluating TOOLFUZZ-AutoFix**   As before, we use the ReAct agent paradigm with the LLM GPT-4o (OpenAI et al., 2024) and temperature set to 0. The results are reported in Table 2. We see that TF-AF outperforms the original documentation on both benchmakrs and all tool significantly.

**Comparison with DRAFT**   We compare TF-AF with DRAFT (Qu et al., 2024), a recent method for improving tool documentation. We note that TOOLFUZZ focuses on error detection, while DRAFT

performs documentation improvement without explicit error detection. We use documentation fixing as a proxy for the TOOLFUZZ's utility. As DRAFT had issues with a subset of tools, we only compare to the tools that DRAFT was able to fix. We see in Table 2 that TF-AF reaches a higher accuracy in all settings, compared to DRAFT. Further, we apply DRAFT to the documentation fixed by TF-AF (TF-AF + DRAFT) to evaluate if DRAFT can further improve documentation. Here we see that DRAFT even degrades the performance of the documentation.

To further investigate this, we compare the unique errors found by TOOLFUZZ before and after DRAFT's documentation fixing. The results are shown in Table 3. We see that while DRAFT is able to reduce the number of unique errors found by TOOLFUZZ on LangChain, it is not able to fix a sizable number of errors on Composio. This indicates that DRAFT might not be able to deliver good performance on tools with good documentation.

Table 3: Unique runtime errors found for: ReACT (RA), ReACT after DRAFT (RA-D),OpenAI Functions (OAF) and Function Calling (FC)

|  | RA | RA-D | OAF | FC |
|---|---|---|---|---|
| Langchain | 51 | 28 | 49 | 54 |
| Composio | 57 | 53 | 52 | 54 |

### 5.4 DIFFERENT AGENT PARADIGMS, TEMPERATURE, AND MODELS

We show that TOOLFUZZ works well across different agent paradigms, LLMs, and temperatures. Specifically, we show in Table 3 TOOLFUZZ's performance accross different agent paradigms, including OpenAI functions and LangChain's function calling (Chase, 2022). We further find, that the low temperature works better for Composio tools, while the high temperature works better for LangChain tools, indicating that depending on the documentations quality, the temperature should be adjusted App. H.1, where we also show that TOOLFUZZ works well across different LLMs.

### 5.5 CASE STUDY

We present a case study of LangChain's pubmed, with two further studies in App. I. The tool's documentation is brief (Figure 9.A). We now consider two erroneous prompts discovered by TOOLFUZZ:

1. Can you list some papers for cancer from 2020 using RNA technology?
2. What are the latest research findings on cancer treatment?

Both prompts concern date handling: the first requests papers from 2020, while the second seeks the latest research. In the first case, pubmed outputs: *"Published: 2024-09-15 Title: A novel small molecule ZYZ384..."*, which is clearly not from 2020. This error occurs because the tool invocation is: pubmed(query="cancer RNA technology 2020"), and this query formulation only searches in paper titles. The PubMed Web-API specifies a particular syntax for querying specific fields, e.g., publication date: ("2020/01/01"[Date - Publication] : "2021/01/01"[Date - Publication]). However, this crucial detail is missing from the LangChain tool documentation, causing the error detected by TOOLFUZZ. We resolved this issue by updating the documentation (see Figure 9.E in App. I). This enables the agent to now formulate an accurate query, resulting in: pubmed(query="cancer AND RNA technology AND ("2020/01/01"[Date - Publication] : "2020/12/31"[Date - Publication])"), which now produces correct output "Published: 2020-11-26 Title: TUG1 long non-coding RNA...".

## 6 CONCLUSION

We introduced TOOLFUZZ, the first method to test tools when used by agents. TOOLFUZZ tests systematically and automatically by building a fuzzing inspired method, integrated with LLM-based prompt generation. We demonstrate TOOLFUZZ's effectiveness in a series of experiments, and show that TOOLFUZZ detects more than 50% more errors at a significantly lower False Discovery Rate. We further introduce two novel benchmarks, focusing on GitHub and file management tasks, to validate the utility of the errors found by TOOLFUZZ and find they can be used to improve the accuracy on the benchmarks significantly in an automated way. This work opens various research directions, including the expansion of TOOLFUZZ to test multiple tools simultaneously to allow the discovery of cross-coordination failures or automated documentation refinement. TOOLFUZZ closes a critical gap in the testing frameworks for LLM-based agents, enabling a more robust and reliable tooling ecosystems.

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

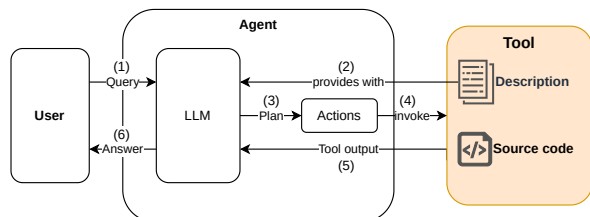

Figure 4: High-level system diagram of an LLM (AI) Agent. The flow of the diagram follows the numbering: (1) User sends a query to the Agent. Alongside with the user query the Agent is provided with the description of all tools available to it (2). With this information the LLM plans (3) actions. Some of these actions require tool invocations (4). After tool calls, an observation is made (5). Based on the observation the agent responds with an answer to the User (6).

## A  BROADER IMPACT

This paper presents work whose goal is to advance the field of Machine Learning. There are many potential societal consequences of our work, none which we feel must be specifically highlighted here.

## B  AGENT

In this work, the primary focus in on how LLM Agents utilize the available tools. This said we consider a minimal agent system design - one containing LLM for reasoning and planning, and tools. This design choice is made to minimize system overhead. The general framework considered can be seen in Figure 4.

## C  SOURCE CODE

The source code for TOOLFUZZ as well as the evalation of TOOLFUZZ is attached as a zip file alongside the paper.

All the instructions on how to run the code base can be found in the README.md file, located at the root directory of the project. The codebase is written in Python 3.10 using a Conda virtual environment. TOOLFUZZ has been developed, run and evaluated on Ubuntu 22.04. All dependencies are listed under the requirements.yml file – instruction on how to setup the environment are also provided in the README.md file.

## D  EXPERIMENTAL SETUP

All of the experiments are run on a single machine without the need of GPU. The used machine for evaluation is equiped with Intel(R) Xeon(R) Platinum 8358 CPU @ 2.60GHz with 30 cores and 222 GB of RAM. Further all experiments are run with budgeting. The selected budget is 5 minutes per tool as a hard limit and a soft token budget of 25 000 tokens. Further we also have a soft price budget of 0.1 USD per tool. Budgets for tokens and price are soft as they would not stop the test execution but would be logged and reported. The variaty of tools is large and some of the tools would go over the tokena and price limits, however most of them are within the limits.

## E  TESTED TOOLS

In this section we present the tools used for the evaluation of TOOLFUZZ. For evaluation of TOOLFUZZ we have used 56 tools from the LangChain library and 83 tools from Composio. The selection criteria for the tools is that they have to be free to use and do not require any API Key or any other form of authentication. We have made exception for the GitHub toolkit as it is widely used. The full list of tools for composio can be seen in Figure 6 and for LangChain in Figure 5.

| | |
|---|---|
| duckduckgo_results_json | requests_post |
| duckduckgo_search | requests_patch |
| youtube_search | requests_put |
| Dall-E-Image-Generator | requests_delete |
| wikipedia | Get_Issues |
| terminal | Get_Issue |
| semanticscholar | Comment_on_Issue |
| Wikidata | List_open_pull_requests__PRs_ |
| python_repl | Get_Pull_Request |
| open-street-map-route-distance | Overview_of_files_included_in_PR |
| pub_med | Create_Pull_Request |
| open-street-map-search | List_Pull_Requests_Files |
| ionic_commerce_shopping_tool | Create_File |
| stack_exchange | Read_File |
| query_graphql | Update_File |
| arxiv | Delete_File |
| Search_NASA_Image_and_Video_Library_media | Overview_of_existing_files_in_Main_branch |
| Get_NASA_Image_and_Video_Library_media_metadata_manifest | Overview_of_files_in_current_working_branch |
| Get_NASA_Image_and_Video_Library_media_metadata_location | List_branches_in_this_repository |
| Get_NASA_Image_and_Video_Library_video_captions_location | Set_active_branch |
| copy_file | Create_a_new_branch |
| file_delete | Get_files_from_a_directory |
| file_search | Search_issues_and_pull_requests |
| move_file | Search_code |
| read_file | Create_review_request |
| write_file | json_spec_list_keys |
| list_directory | json_spec_get_value |
| requests_get | python_repl_ast |

Figure 5: List of Langchain Tools used for the evaluation.

EMBED_TOOL_CREATE_IMAGE_VECTOR_STORE

WEBTOOL_SCRAPE_WEBSITE_CONTENT

FILETOOL_GIT_REPO_TREE

FILETOOL_WRITE

BROWSER_TOOL_SCROLL_PAGE

computer

WORKSPACE_TOOL_WORKSPACE_STATUS_ACTION

RAGTOOL_ADD_CONTENT_TO_RAG_TOOL

SHELLTOOL_CREATE_SHELL

FILETOOL_OPEN_FILE

FILETOOL_RENAME_FILE

BROWSER_TOOL_CLICK_ELEMENT

WEBTOOL_SCRAPE_WEBSITE_ELEMENT

GIT_GITHUB_CLONE_CMD

FILETOOL_CREATE_FILE

FILETOOL_SEARCH_WORD

ZEPTOOL_CREATE_SESSION

FILETOOL_FIND_FILE

MATHEMATICAL_CALCULATOR

BROWSER_TOOL_TYPE_TEXT

FILETOOL_LIST_FILES

CODE_ANALYSIS_TOOL_GET_RELEVANT_CODE

ZEPTOOL_ADD_MEMORY

SHELLTOOL_SPAWN_PROCESS

SHELLTOOL_TEST_COMMAND

FILETOOL_GIT_CUSTOM

ZEPTOOL_GET_MEMORY

GIT_GET_PATCH_CMD

FILETOOL_SCROLL

ZEPTOOL_SEARCH_MEMORY

SQLTOOL_SQL_QUERY

HISTORY_FETCHER_GET_WORKSPACE_HISTORY

CODE_ANALYSIS_TOOL_CREATE_CODE_MAP

CODE_ANALYSIS_TOOL_GET_METHOD_SIGNATURE

FILETOOL_GIT_CLONE

RAGTOOL_RAG_TOOL_QUERY

CODE_FORMAT_TOOL_FORMAT_AND_LINT_CODEBASE

EMBED_TOOL_QUERY_IMAGE_VECTOR_STORE

FILETOOL_GIT_PATCH

SPIDERTOOL_CRAWL

CODE_ANALYSIS_TOOL_GET_CLASS_INFO

FILETOOL_CHANGE_WORKING_DIRECTORY

GIT_GIT_REPO_TREE

BROWSER_TOOL_GET_SCREENSHOT

IMAGE_ANALYSER_ANALYSE

CODE_ANALYSIS_TOOL_GET_METHOD_BODY

SHELLTOOL_EXEC_COMMAND

GREPTILE_CODE_QUERY

SPIDERTOOL_SCRAPE

BROWSER_TOOL_GET_ELEMENT_DETAILS

str_replace_editor

BROWSER_TOOL_NAVIGATE_HISTORY

BROWSER_TOOL_GOTO_PAGE

BROWSER_TOOL_REFRESH_PAGE

FILETOOL_EDIT_FILE

BROWSER_TOOL_GET_PAGE_DETAILS

COMPOSIO_WAIT_FOR_CONNECTION

HACKERNEWS_GET_USER

COMPOSIO_RETRIEVE_ACTIONS

CODEINTERPRETER_EXECUTE_CODE

HACKERNEWS_GET_ITEM_WITH_ID

COMPOSIO_GET_RESPONSE_SCHEMA

COMPOSIO_ADVANCED_USE_CASE_SEARCH

COMPOSIO_INITIATE_CONNECTION

WEATHERMAP_WEATHER

COMPOSIO_GET_REQUIRED_PARAMETERS

COMPOSIO_EXECUTE_ACTION

COMPOSIO_ENABLE_TRIGGER

ENTELLIGENCE_INTERACT_WITH_THE_REPOSITORY_AGENT

CODEINTERPRETER_RUN_TERMINAL_CMD

HACKERNEWS_SEARCH_POSTS

CODEINTERPRETER_GET_FILE_CMD

COMPOSIO_CHECK_ACTIVE_CONNECTION

HACKERNEWS_GET_FRONTPAGE

HACKERNEWS_GET_LATEST_POSTS

COMPOSIO_SEARCH_DUCK_DUCK_GO_SEARCH

HACKERNEWS_GET_TODAYS_POSTS

TEXT_TO_PDF_CONVERT_TEXT_TO_PDF

COMPOSIO_LIST_TRIGGERS

COMPOSIO_LIST_APPS

ENTELLIGENCE_ADD_A_NEW_REPOSITORY

COMPOSIO_RETRIEVE_APPS

CODEINTERPRETER_UPLOAD_FILE_CMD

Figure 6: List of Composio Tools used for the evaluation.

## F  BENCHMARKS

Table 4: Binomial Confidence Intervals at 95% confidence for Table 2. TOOLFUZZ Auto Fix (TF-AF), DRAFT and DRAFT applied on top of TOOLFUZZ Auto Fix for the File Management (FMB) and GitHub Benchmarks (GHB)

|  | Tool | Original | TF-AF | DRAFT | TF-AF + DRAFT |
|---|---|---|---|---|---|
| FMB | Terminal | [0.25, 0.36] | [0.34, 0.46] | [0.33, 0.45] | [0.27, 0.38] |
|  | FileToolkit | [0.28, 0.40] | [0.35, 0.47] | [0.30, 0.42] | [0.28, 0.39] |
| GHB | GithubToolkit | [0.13, 0.37] | [0.34, 0.46] | [0.18, 0.43] | [0.15, 0.39] |

For both benchmarks, we have used the same autofix setup. In both cases GPT-4o is utilized with the following prompts: Figure 22, Figure 24, Figure 23.

Further we generated 10 fixes for each tool using the TOOLFUZZ's Auto Fix (TF-AF) and DRAFT. Then the generated fixes are evaluated using the validation split of the benchmark, to remove selection bias. The best performing fix is then selected for the full benchmark evaluation.

### F.1  FILE MANAGEMENT TOOLKIT BENCHMARK

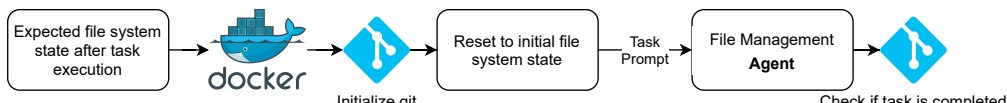

Figure 7: File management benchmark setup

The file management toolkit benchmark consists of 42 domain-specific environments, which are listed in Figure 8.

Each task of the benchmark is setup in a Docker container. The Docker container has the initial state of the file system as well as a ReAct agent with the tool under test. The agent is then presented with the task prompt in the initialized environment. Upon completion of the agent's execution, the success of the task is verified using a `git diff` between the initial and final states (Figure 7).

Further we have computed the binomial confidence intervals for the results from Table 2 in Table 4. The confidence intervals clearly show that the TOOLFUZZ's Auto Fix (TF-AF) and DRAFT are able to improve the performance of the tools. Further the confidence intervals again show that TOOLFUZZ is slightly better than DRAFT.

### F.2  GITHUB TOOLKIT BENCHMARK

The GitHub toolkit benchmark consists of 18 Comment Issues tasks, 9 Update File tasks, 2 Create Pull Requests tasks, 6 Create File tasks, 9 delete file tasks and 10 Create branch tasks. For each of these tasks, we have a script that reverts the state of the repository to the initial states and a script which verifies the success of the tasks.

Again we have computed the binomial confidence intervals for the results from Table 2 in Table 4. The confidence intervals clearly show that the TOOLFUZZ's Auto Fix (TF-AF) and DRAFT are able to improve the performance of the tools. Further the confidence intervals again show that TOOLFUZZ is better than DRAFT.

## G  MANUAL LABELING

The runtime failure detection cannot suffer from false positives as the tool is either throwing a runtime error or not. However for correctness detection, there is no ground truth or detemenistic way to evaluate the correctness of the tool. That is why TOOLFUZZ uses both an LLM evaluation and Input/Output consistency checks. In order to evaluate our approach we need to aquire a ground truth.

| | | |
|---|---|---|
| Accountant | Finance and Banking | Machine Learning |
| Agriculture and Precision Farming | Forensic Science | Manufacturing and Automation |
| Art | Geology and Geophysics | Meteorology and Climate Science |
| Artificial Intelligence | Government and Public Administration | |
| Astronomy and Space Exploration | Graphic Design and Animation | Music Production and Sound Engineering |
| Biotechnology | Healthcare and Medicine | Pharmaceuticals and Drug Development |
| Computer Science | History and Archival Science | |
| Construction and Architecture | | Psychology and Neuroscience |
| Cybersecurity | Hospitality and Tourism | Real Estate |
| Data Science | Human Resources and Recruitment | Robotics |
| E-commerce and Retail | Journalism and Digital Media | Social Media and Digital Marketing |
| Economics and Market Research | Law and Legal Analytics | Sports Science and Analytics |
| Education and E-learning | Linguistics and Natural Language Processing | |
| Energy and Utilities | Logistics and Supply Chain Management | Telecommunications |
| Entertainment and Media | | Video Game Development |
| Environmental Science | Logistics Tour | |

Figure 8: List of environment domains with file systems and tasks for the File Management Domain

For this we had to manually label the correctness of the tool invocations. The labeling process was done by the authors of this work. The labeling process was done in two steps. As manual labeling is labourous we had to subsample from the full run. In the case of the baselines we sampled 10 prompts per tool, while in the case of TOOLFUZZ we sampled 10 prompt sets per tool as effectively one prompt set corresponds to one error in the case of TOOLFUZZ and one prompt corresponds to one error for the baseline. To remove bias in while labeling only the generated user query, the tool input and output, and the agent response are visible to the labeler. Then the labeler labels the trace as erroneous or not.

## H  ADDITIONAL EVALUATION

To accompany the evaluation presented in the main paper, we have conducted additional experiments to further evaluate the effectiveness of TOOLFUZZ.

### H.1  LLM'S IMPACT ON TOOLFUZZ

Table 5: Number of unique runtime errors using different LLMs for the agent and the TOOLFUZZ's prompt generation. Rows - Agent LLMs, Columns TOOLFUZZ's prompt LLMs.

| | GPT-4o-mini | GPT-4o | Claude 3.5 Haiku | Gemini-2.0-flash |
|---|---|---|---|---|
| GPT-4o-mini | 167 | 165 | 166 | 154 |
| GPT-4o | 143 | 142 | 123 | 131 |
| Claude 3.5 Haiku | 93 | 117 | 99 | 109 |
| Gemini-2.0-flash | 83 | 90 | 81 | 106 |

In this section, we present the results of the evaluation of TOOLFUZZ using different LLMs. The results are shown in Table 5. The table shows that results vary depending on the used model for both the agent and TOOLFUZZ, however always finding errors. This indicates that TOOLFUZZ is

Table 6: Evaluating unique error finding performance with varying LLM temperature for prompt generation

|  | t=0 | t=0.4 | t=0.8 | t= 1 |
|---|---|---|---|---|
| Langchain | 84 | 82 | 98 | 90 |
| Composio | 78 | 61 | 69 | 56 |

robust and can be used with different LLMs without significant impact on its performance. Further since LLMs are black boxes and many of the tools are using closed sourced APIs in many cases they cannot be changed which leaves space for only tool description optimization.

Further we investigate the impact of different model parameters on the performance of TOOLFUZZ. The results are shown in Table 6. The table shows that a lower temperature works better for well crafted tools like the ones from Composio (Composio, 2025), as they are more strict and would reject more creative prompts. While a higher temperature leads to more diverse and creative prompts which can showcase a lot of weaknesses in weakly defined tools like the ones from LangChain (Chase, 2022).

## H.2 STATISTICAL SIGNIFICANCE

Table 7: Binomial Proportional Confidence Intervals for the FDR (%) for Table 1

|  | Baseline W | Baseline G | TOOLFUZZ CC | TOOLFUZZ LO | TOOLFUZZ |
|---|---|---|---|---|---|
| Langchain | $0.676_{\pm 0.075}$ | $0.845_{\pm 0.038}$ | $0.426_{\pm 0.057}$ | $0.364_{\pm 0.053}$ | $0.153_{\pm 0.05}$ |
| Composio | $0.805_{\pm 0.035}$ | $0.789_{\pm 0.051}$ | $0.483_{\pm 0.053}$ | $0.472_{\pm 0.050}$ | $0.374_{\pm 0.066}$ |

Table 8: Total unique runtime errors TOOLFUZZ can find estimated by the standard Chao1 estimator - 95% confidence interval for: ReACT (RA), ReACT after DRAFT (RA-D),OpenAI Functions (OAF) and Function Calling (FC)

|  | RA | RA-D | OAF | FC |
|---|---|---|---|---|
| Langchain | [51 - 54] | [28 - 36] | [50 - 57] | [58 - 72] |
| Composio | [61 - 84] | [89 - 175] | [54 - 77] | [64 - 108] |

In this section, we present the confidence intervals (Table 7) for the results present for the correctness evaluation in Table 1. It can be seen from the table that TOOLFUZZ outperforms the baselines in all cases, even considering the lower bound of the confidence interval with the upper bound of the baseline.

To show the power of TOOLFUZZ to find unique errors, we present the results of the unique errors found by TOOLFUZZ and DRAFT in Table 3. We can see that TOOLFUZZ is able to find unique errors regardless of the agent used and also on top of DRAFT's improved tool descriptions. To further strengthen this claim, we present the number of total unique errors estimated by the standard Chao1 estimator (Chao, 1984; Colwell, 2009) and report its 95% confidence interval (TUE-CI). The results are shown in Table 8. The table again shows that regardless of the agent or DRAFT's fix there are still unique errors to be found.

## H.3 CROSS TOOL CALLING

We have used prompts generated from TOOLFUZZ to assess whether tool documentations are too broad, leading to unintended activations across different categories. As discussed in §2, under-specification can prompt undesired tool invocations. We categorized tools by their domain and sampled prompts intended for different tool groups. Our experiment revealed that 492 prompts led to unplanned tool usages across 53 tools in langhcain. Detailed results can be found in the experiment folder within the source code (./src/eval/cross_tool_calling). The tool categories are presented in the paragraph below.

**Tools categorization**  As we wanted to test cross tool calling, we have grouped the tools listed in App. E into groups with respect to their domain. The groups are as follows:

1. Dall-E: Dall-E-Image-Generator

2. NASA Image and Video Library: Get Nasa Image and Video Library media metadata manifest, Get NASA Image and Video Library media metadata location, Search NASA Image and Video Library media, Get NASA Image and Video Library video captions location

3. Knowledge Repositories: wikidata, wikipedia, stack_exchange

4. Academic Resources: arxiv, semanticscholar, pubmed

5. Search engines: duckduckgosearch, duckduckgosearchresult, stack_exchange, youtube_search

6. File operations: file_search, list_directories, terminal, python_repl, python_repl_ast

7. File deletions: file_delete, terminal, python_repl, python_repl_ast

8. Move files: move_file, terminal, python_repl, python_repl_ast

9. Read files: read_file, terminal, python_repl, python_repl_ast

10. Copy files: copy_file, terminal, python_repl, python_repl_ast

11. HTTP Requests: request_delete, terminal, python_repl, python_repl_ast, requests_get, requests_patch, requests_post, requests_put

12. Map search: open-street-map-search

13. JSON Operations: json_spec_list_keys, json_spec_get_value

14. Directions: open-street-map-distance

15. GraphQL: query_grapql

## I  CASE STUDIES

In this section, we present two additional case studies in addition to the one presented in the main article (§5.5). The tools we are looking at are `open-street-map-directions`, tool for navigating on open street map which is developed by the authors and a LangChain tool `terminal` which is designed to execute shell commands on the host system.

**OpenStreetMap directions**  Now let's look at an example of the open street map. Here are the prompts in the prompt set $\mathbb{P}$:
1. What is the car route distance from Paris to Lyon?
2. What is the car route distance from Paris, France to Lyon, France?
3. What is the car route distance from City of Paris to City of Lyon?
4. What is the car route distance from Paris FR to Lyon FR?
5. What is the car route distance from Paris France to Lyon France?

For this prompt set the generated expected answer is: *"The road distance between Paris and Lyon is approximately 465 kilometers (289 miles)."*.

After invoking the agent with those prompts, there are 2 output buckets. The first bucket contains: 1, 2, 4 and 5 with tool output – *"The distance between City of Paris to Lyon is 939224.1 meters"*, while the second is just one prompt 3 with output: *"The distance between Paris to Lyon is 465460.3 meters"*.

The input argument buckets are three, grouped as follows:

- bucket 1: {1} with parameters:
  {from_location_query:  "Paris", to_location_query:  "Lyon"}
- bucket 2: {2, 4, 5} with parameters:
  {from_location_query:  "Paris, France", to_location_query:  "Lyon, France"}

**A**

**Original tool documentation**
  **Tool function declaration:** def open_street_map_(query: str) → str:
  **Tool documentation:** *Tool to query a map. This tool can locate places by name and simple queries such as: libraries in San Francisco. The idea of the tool is to locate places like coffee shops, offices, etc.*

**B**

**Original PubMed documentation**:
  **Input arguments:** query - type string.
  **Tool documentation:** *A wrapper around PubMed. Useful for when you need to answer questions about medicine, health, and biomedical topics from biomedical literature MEDLINE, life science journals, and online books. Input should be a search query.*

**C**

**Original Terminal tool documentation**:
  **Input arguments:** commands - type: Union[str, List[str]], description: List of shell commands to run. Deserialized using json.loads.
  **Tool documentation:** *Run shell commands on this Linux machine.*

**D**

**Fixed tool documentation**
**Tool function declaration:** def open_street_map_(query: str) → str:
**Tool documentation:** *Tool which can find a route between two locations and give back the distance in km of that route. The route is on roads that can be driven with a car. The tool provides route distance in km for a car trip between the two locations. The two locations can be cities or concrete places i.e. office buildings, shops, parks, and so on. **Tool arguments with cities must always include the full names and countries i.e. NYC → New York City, USA or Paris → Paris, France.***

**E**

**Fixed PubMed documentation**:
  **Input arguments:** query - type string.
  **Tool documentation:** *A wrapper around PubMed. Useful for when you need to answer questions about medicine, health, and biomedical topics from biomedical literature, MEDLINE, life science journals, and online books. Input should be a search query. **The query has special syntax for different fields in the paper:***
*The list of available search fields is: All Fields, Author, Date - Create, Date - Publication, EC/RN Number, Editor, Title, Title/Abstract, Transliterated Title, VolumeFor some fields i.e. the date fields ranges are available. Here is an example of a date query and its usage.*
*List papers that are from 2010: ("2010/01/01"[Date - Entry] : "2011/01/01"[Date - Entry])*
*Most of the other fields are used as follows:*
*List papers with first author John Doe: John Doe[Author].*
*In order to combine a filter with multiple fields we use AND. Here is an example:*
*What are the papers with Last Author Ivan from 2020 until now:(Ivan[Author - Last]) AND (("2020/02"[Date - Create] : "3000"[Date - Create]))*

**F**

**Fixed Terminal tool documentation**:
  **Input arguments:** commands - type: Union[str, List[str]], description: List of shell commands to run. Deserialized using json.loads.
  **Tool documentation:** *Execute shell commands on this Linux machine. Use clear, concise commands with relative paths. Enclose paths with spaces in quotes. Validate command syntax and file paths. The tool can be invoked multiple times to ensure correct file system state.*

Figure 9: a case study comparing the original and improved descriptions of three tools: open_street_map, pubmed, and terminal. The first set of three red boxes represents the original descriptions of these tools, which are prone to failures, while the second set of three green boxes shows the same tools with improved descriptions designed to reduce errors.

- bucket 3: {3} with input parameters:
    {from_location_query: "City of Paris", to_location_query: "City of Lyon"}

From the bucketing, we can make the observation that the inputs with full country name work well, and using just a city name works half the time (input bucket 1 and bucket 3) so one way to fix this is to add the requirement of country name in the argument to the tool's description. The fixed description can be found in Figure 9.D. With the fixed description, the example prompt set passes.

**Terminal** The terminal tool has a very short and generic description as shown in Figure 9.C. This description, however, does not explain at all how to properly use the tool i.e. in what way the command parameters have to be presented, which commands can be used etc.

Here is an erroneous set of prompts found by TOOLFUZZ:

1. How big is the assignment_1_draft.docx file in the Contract Law assignments folder?

2. Find the size of the draft of assignment 1 file in the /[FULL_PATH]/Law_Student_Files/Academics/Courses/Semester_1/... directory.

> Please invoke the {tool.name} exactly like: {tool.name}({bad_arg})

Figure 10: Sanity check prompt for forcing the agent to invoke the tool under test with the erroneous arguments.

3. What is the file size of the first draft of assignment 1 in the Contract Law assignments folder?

4. Find the size of the assignment one draft file in the /[FULL_PATH]/Law_Student_Files/Academics/Courses/Semester_1/... directory.

The FULL_PATH is in fact the full path; however, we have masked it out as it is part of the machine's file structure.

The prompt set has the goal of finding the size of the first Assignment in the Contract law assignments folder. It is important to note that the agent is unaware of the directory in which it is running.

In this case, both the number of input and output buckets is four – each prompt resulted in a unique input parameters and tool output:

1. tool input: {"commands":"du -h /Contract Law/assignments/assignment_1_draft.docx"}
   tool output: cannot access '/Contract Law/assignments/assignment_1_draft.docx': No such file or directory

2. tool input: {"commands":"du -sh /[FULL_PATH]/Assignments/assignment_1_draft.*"}
   tool output: 0 /[FULL_PATH]/assignment_1_draft.docx

3. tool input: {"commands":"ls -lh ~/Contract Law/assignments/ | grep 'assignment 1 draft'"}
   tool output: "ls: cannot access '~/Contract Law/': No such file or directory"

4. tool input: {"commands":"ls [FULL_PATH]/Law_Student_Files/Templates"}
   tool output: ""

As can be seen, each prompt results in unique input parameters and tool output. We can see that twice in prompts 1 and 3 the tool tries to access invalid paths. For prompt 4 the path is correct; however, the task at hand is not just to list a directory. The general error here seems to be related to the provided paths – the description does not give any inside information to the agent on where the Terminal tool is or how to navigate. With this information found by TOOLFUZZ and using the previously described autofix method (§5.3) a new description is generated (Figure 9). With this new description, the prompt set is now valid by just using ls multiple times to properly navigate the file system.

## J TOOLFUZZ PROMPTS

In this subsection of the Appendix all prompts used for TOOLFUZZ are listed. For all the following prompts, string templates are used – {variable} will be inserted with the corresponding variable when the prompt is constructed.

### J.1 TOOLFUZZ RUNTIME TOOL FAILURE DETECTION

For the runtime error detection as explained in §4 the heavy lifting is done by the Fuzzer so an LLM is leveraged only for sanity check with the prompt Figure 10 and for converting the erroneous argument into a user query which is done with Figure 11.

### J.2 TOOLFUZZ CORRECTNESS DETECTION

**Prompt set generation** The generation of prompt sets as described in §4.2 involves multiple LLM generations. Firstly, the prompt template is generated given the tool description and in some cases additional context, the full prompt is given in Figure 12

> You are an AI assistant tester. The idea is to come up with prompts which will make the following tool produce incorrect answer.
> Tool information:{tool_information}
> Please generate prompts which will make the following tool produce incorrect answer.
> {format_instructions}

Figure 11: Prompt for generating realistic user prompt which will invoke the tool under test with erronoeous arguments.

> You are a user asking an AI assistant for help - just speak naturally.
> Your task:
> Generate **template questions** that a user might ask based on a tool's purpose and capabilities.
> The tool is described as follows:
> {tool_prompt}
> With the following context:
> {tool_context}
> Here are some examples for inspiration:
> For a map/distance tool:
> What is the distance from [A] to [B]?
> How much time would it take to go from [A] to [B]?
> If I start from [A] and go to [B] with [C] km/h average speed how much time would it take me?
> For a knowledge based or information retreieval tool:
> What do you know about [A]?
> What is/are [A] for [B]?
> Is it true that [A] is [B]?
> Is [A] related to [B]?
> In what year did [A] happen?
> When was [A] born?
> Find [A] in [B]?
> [A] my work to [B].
> [A] from [B] to [C].
> Also some more specific questions like:
> Is it true that Mr [A] was related to Mrs. [B]?
> I am at [A], how much time it will take me to go from the closest [B] to the [C] airport?
> Find an article/paper/document written by [A] on topic [B]?
> Use just words within the placeholder brackets i.e. [A], [B], [C], [destination], [source], [topic], [place], [location], [person], [company], [organization], [event], [date] etc. Avoid adding any special characters or punctuation marks AVOID [locations_rating] or [bracket_selector[]].
> Be creative, and make sure the templates match how people might actually talk to an AI assistant which has this tool i.e. both specific templates and more general ones.
>
> {format_instructions}.

Figure 12: Prompt for generating template questions/prompts for the agent, given the tool under test description and additional context if needed.

The next step is to infill the generated template questions/prompts with synonymous phrases. For this, the following prompt is used to generate the phrases which are later inserted in the string templates Figure 13.

As in some cases when giving context to the prompt generation, some facts are taken directly from the context which is not very human, so we have also introduced a humanizing prompt Figure 14.

## J.3 LLM ORACLE

The LLM Oracle has to evaluate the Agent answer according to an expectation. Firstly, the expectation is generated.

The LLM expectation generation is again a multistage process. Firstly, an LLM is prompted to answer to each of the generated prompts from a set of prompts $p \in \mathbb{P}$ – Figure 15. After all the answers are generated, we use them to prompt again an LLM to come up with the LLM expectation – Figure 16.

> You are a user interacting with an AI assistant—just speak naturally.
> Your task:
> For the following template question:
> '{template_prompt}'
> Generate infills for each placeholder (like [A], [B], etc.).
> Each infill should be a **specific example** that could realistically fill that blank — such as named entities, objects, or well-known terms. These should not be rephrasings of the whole sentence or template. Infills should reflect how real users might refer to the same thing using:
> - Synonyms
> - Abbreviations
> - Rephrasings
> - Alternate spellings
> - Common names or titles
> Use only inputs that are **relevant** to this tool:
> {tool_prompt}
> And the tool's background or purpose givent the following context: {tool_context}
> **Examples**:
> Template: 'What are some [A] in [B]?'
> Infills for A: ['Coffee Shop', 'Cafeteria', 'Coffeehouse', 'Café']
> Infills for B: ['Zurich', 'ZH', 'Zurich Switzerland', 'ZH CH', 'ZH Switzerland', 'Zurich CH']
> Template: 'Who is [A]?'
> Infills for A: ['Albert Einstein', 'A. Einstein', 'Alb. Einstein', 'Einstein']
> Template: 'When did [A] happen?'
> Infills for A: ['World War I', 'World War One', 'WW 1', 'First World War']
> Template: 'What are the latest news in [A]?
> Infills for A: ['cinema', 'hollywood', 'kino', 'movies', 'show business']
> Template: 'What are the [A] in [B]?'
> Infills for A: ['latest news', 'current events', 'breaking news', 'daily news', 'daily events']
> Infills for B: ['politics', 'government', 'public affairs']
> Template: 'I am at [A], how much time it will take me to go from the closest [B] to the [C]?'
> Infills for A: ['Zurich HB', 'Zurich main train station', 'Zurich main station']
> Infills for B: ['Mc Donalds', 'fast food restaurant McDonalds', 'McD burgers']
> Infills for C: ['ETH HG Bibliothek', 'ETH main building library', 'ETH main library']
> Template: 'Can you find [A] in [B]?'
> Infills for A: ['family picture', 'png with the family', 'family photo', 'family portrait']
> Infills for B: ['the home directory', 'my workspace', 'main directory']
> Template: '[A] [B] to [C]'
> Infills for A: ['Submit', 'Send', 'Upload', 'Commit']
> Infills for B: ['main.py', 'the main python file', 'src/main', 'the main source file']
> Infills for C: ['the server', 'the cloud', 'the repository', 'the remote branch']
> Template: '[A] my work to [B]'
> Infills for A: ['Move', 'Transfer', 'Cut']
> Infills for B: ['archive folder', 'the archive']
> **INVALID EXAMPLES:**
> Template: 'What are the side effects of [medication] according to recent studies?'
> Infills for A: ['medication', 'this medicine', 'this therapeutic']
> **Important:**
> Do **NOT** reuse any examples from the following list: {used_args}.
> {format_instructions}.

Figure 13: Prompt for generating infills for the masks in the prompt templates.

Lastly, the LLM oracle has to evaluate the Agent output, for this prompt we are using both reasoning and scoring between 1 and 10 as previous experiments showed that making the LLM to just evaluate with yes/no gave more false positives – Figure 17.

### J.4 BASELINE PROMPTS

For the baselines from §5.2. We use the following prompt to generate test prompts - Figure 18 and Figure 19. Here, the main difference is that in the white box scenario the tool_info variable will contain both the tool description and the tool source code while in the gray box the tool source code is not included.

> Given the following tool description: '{tool_prompt}' and the following tool prompts that are synonymous: '{prompts}' Please make such that the prompts are like a person would write it and not a machine, so nothing too concrete but also not too vague.
> {format_instructions}

Figure 14: Prompt for making more human like prompts. In some cases prompts are too specific i.e. full paths or full identificators which users might shorten or write more intuitively.

> You are emulating the following tool: {tool_prompt}. Given the tool return value for the following questions: {questions}
> Example:
> Tool description: Tool which can find a route between two locations and give back the distance in km of that route. The route is on rodes that can be driven with car. The tool provides route distance in km for car trip between the two locations.
> The two locations can be cities or concrete places i.e. office buildings, shops, parks and so on.
>
> Questions:
> What is the distance between Sofia and Zurich?
> What is the distance between SF and ZH?
> What is the distance between Sofia BG and Zurich CH?
> What is the distance between Sofia Bulgaria and Zurich Switzerland?
>
> Answers:
> The road distance between Sofia, Bulgaria, and Zurich, Switzerland is approximately 1,450 kilometers (900 miles).
> If "SF" refers to San Francisco (SF), California, USA and "ZH" refers to Zurich (ZH), Switzerland, the distance is vast. Since it's impossible to drive directly due to the Atlantic Ocean, the driving distance would be irrelevant. However, hypothetically driving the distance across land would exceed 9,500 kilometers (5,900 miles).
> "BG" stands for Bulgaria and "CH" stands for Switzerland. The road distance between Sofia, Bulgaria, and Zurich, Switzerland is approximately 1,450 kilometers (900 miles)
> The road distance between Sofia, Bulgaria, and Zurich, Switzerland is about 1,450 kilometers (900 miles).
> {format_instructions}.

Figure 15: Prompt for generating answers for given question. Used for expectation generation for the LLM Oracle.

In addition to these prompts, the baseline also has their own LLM judge using the following prompt for evaluation – Figure 20.

As mentioned in §5 the baselines can fall into degenerate string generation - examples of such cases can be seen in Figure 21.

You have the following sentences: {sentences}
Make once sentence which is factually correct to you and most of the sentences, not required to represent all of the answer but MUST be approximately correct.

Examples:

Sentences:
The road distance between Sofia, Bulgaria, and Zurich, Switzerland is approximately 1,450 kilometers (900 miles).
If "SF" refers to San Francisco (SF), California, USA and "ZH" refers to Zurich (ZH), Switzerland, the distance is vast. Since it's impossible to drive directly due to the Atlantic Ocean, the driving distance would be irrelevant. However, hypothetically driving the distance across land would exceed 9,500 kilometers (5,900 miles).
"BG" stands for Bulgaria and "CH" stands for Switzerland. The road distance between Sofia, Bulgaria, and Zurich, Switzerland is approximately 1,450 kilometers (900 miles)
The road distance between Sofia, Bulgaria, and Zurich, Switzerland is about 1,450 kilometers (900 miles).
Answer: The expected distance is around 1500 kilometers. {format_instructions}.

Figure 16: Prompt for generating the LLM Expectation for particular tool set $\mathbb{P}$

You are assessing an agent RESPONSE. You should give a score between 1 and 10. 10 means that the answer satisfies both criteria: first is the tool output and second is LLM expectation. The agent response should be within 10% margin of error or false otherwise. 0 is for when both criteria are not satisfied.
Alongside the evaluation of true/false, you should provide reasoning for it.

Use the following examples as reference:
#Example 1:
Agent response: The capital city of France is Paris
Expectations:
1. No results found for the query
2. Paris
correctness_degree: 5
Reason: The response doesn't comply with the first expectation: 'No results found for the query'

#Example 2:
Agent response: As of now, the current survey results for the 2024 US presidential election show Kamala Harris leading Donald Trump. Here are the key figures:

- **Kamala Harris**: 46%
- **Donald Trump**: 48%
- **Robert F. Kennedy Jr.**: 5.5%

Harris's popularity has seen a significant increase, with half of voters viewing her favorably, which is an improvement from 43% the previous week.
Expectations:
1. Kamala Harris, Donald Trump.
2. The current survey results for the 2024 US presidential election show candidate A leading with X%, followed by candidate B with Y%, and other candidates with Z% and W%.
correctness_degree: 9
Reason: The response mentions both Kamala Harris and Donald Trump, thus complying with the first expectation, and also presents percentages, thus complying with the second.

#Example 3:
Agent response: If you're looking for a solid, mid-range option, expect to spend around $400 to $500, there is one cheaper option around 350 USD
Expectation:
1. $347.99
2. A TV sells in the range of 400-800 USD
correctness_degree: 8
Reason: The response is close enough to the expected values and is based on the expected tool output.

#Example 4:
Agent response: The distance between LA and Sofia is 1,047 miles
Expectation:
1. Distance from Sofia to LA is 10,633 km
2. The distance between Sofia and LA is 10,000 KM
correctness_degree: 2
Reason: The response doesn't comply with both expectations - the error is too big, but is still mentioning the correct names.

Now please evaluate the following agent response and give the reason for your evaluation:
'{agent_output}'
Expectations:
1. {tool_output}
2. {expected}
{format_instructions}

Figure 17: Prompt used by the LLM Oracle to evaluate the correctness of an agent answer given the tool and expectation as well as the output of the tool under test

You are an AI assistant tester. The idea is to come up with prompts which will make the following tool crash.
Example:
Tool information: This tool can divide two numbers.
Crash prompts: [
"Can you please divide 10 by 0",
"If I have no friends left how can I divide my apples evenly between them?",
"veryyyyyyyy looooooong stringgggggg, veryyyy looooong stringgggg"
]
Tool information: {tool_info}
**DO NOT GENERATE** "a"+"b" or "str".repeat(1000) or similar.
Now generate the crash prompts, you MUST give the **full string prompt** and no further logic has to be ran on these prompts.
Give just the prompts, no need for explanation or reasoning at the end. Just clean json.

Figure 18: Prompt for generating prompts with the intention of triggering runtime tool failures.

You are an AI assistant tester. The idea is to come up with prompts which will make the following tool produce incorrect answer.
Tool information:{tool_information}
Please generate prompts which will make the following tool produce incorrect answer.
{format_instructions}

Figure 19: Prompt for generating prompts for correctness evaluation.

You are an AI assistant judge. Your job is to evaluate and reason on the correctness of the given answer knowing the question is:
{question}
The provided answer is: {answer}

Is the answer correct? Please provide reasoning.
{format_instructions}

Figure 20: The prompt used by the LLM Judge for baseline correctness evaluation

**Degenerate prompt generations**
```
'/home/user/../../../../../../../../../../../../../../../../../../../../../../../../...'
Search for pattern: '*' dir_path: '/var/lib/dpkg/lock-frontend-frontend-frontend-...'
```
**Out of distribution examples:**
Search for the latest recipes in Italian cuisine on arxiv.
Search for the most popular movies of all time on arxiv.
Search for the Garden of Eden.
Find the nearest time machine rental.
Find the nearest Hogwarts School of Witchcraft and Wizardry.
Find the nearest black hole.
Find the population of New York.
Search for the nearest alien spaceship.

Figure 21: Out of distribution and degenerate prompt generation by the baselines

You are an AI Agent Tool developer can you make the description of a tool more precise and clear. The problem is that the LLM sometimes doesn't translate the queries correctly.
This comes from the fact that the tool description doesn't explain what exactly is allowed and how to use the tool correctly.
The current description of the tool is: {tool_description}

Here are some failing examples of the tool in action:
{bad_examples}

Given the bad examples please identify the main issues on those examples and what is the cause of that issue. How can these issues be avoided by validation i.e. with this tool or external resources or it's just a user mistake? List the main issues and how to avoid them.

Now that the issues and how to avoid them are clear. Please create tool description that addresses these issues. The description is a manual on how to use the tool correctly and what is allowed and what is not.
It should explain how to avoid the issues that were found in the examples. Add that the tool can be invoked multiple times for better validation of the file system state.

Also give few examples if you think they are applicable.

Please provide description which reflects these issues. The new description shouldn't be longer than 100 words.

{format_instructions}

Figure 22: Prompt for automatic tool description fixing, based on the tool's description and as well as a set of prompts resulting into tool failures (bad_examples) found by TOOLFUZZ.

You are an AI Agent Tool developer can you make the description of a tool more precise and clear. The problem is that the LLM sometimes doesn´t translate the queries correctly.
This comes from the fact that the tool description doesn´t explain what exactly is allowed and how to use the tool correctly.
The current description of the tool is: {tool_description}

The description has to be a manual on how to use the tool correctly and what is allowed and what is not.
Please provide just the new description of the tool.
{format_instructions}

Figure 23: Prompt for automatic tool description fixing, based only on the tool description.

You are an AI Agent Tool developer can you make the description of a tool more precise and clear. The problem is that the LLM sometimes doesnt́ translate the queries correctly.
This comes from the fact that the tool description doesnt́ explain what exactly is allowed and how to use the tool correctly.
The current description of the tool is: {tool_description}

Here are some failing examples of the tool in action:
{bad_examples}

Given the bad examples please identify the main issues on those examples and what is the cause of that issue. How can these issues be avoided by validation i.e. with this tool or external resources or itś just a user mistake? List the main issues and how to avoid them.

Now that the issues and how to avoid them are clear. Please create tool description that addresses these issues. The description is a manual on how to use the tool correctly and what is allowed and what is not. It should explain how to avoid the issues that were found in the examples. Add that the tool can be invoked multiple times for better validation of the file system state.

Also give few examples if you think they are applicable.

Please provide description which reflects these issues. The new description shouldnt́ be longer than 100 words.

{format_instructions}

Figure 24: Prompt for automatic tool description fixing, based on both the tool's description and tool's source code.

