# OpenReview forum: "ToolFuzz: Automated Agent Tool Testing"
_ICLR.cc/2026/Conference — Submitted to ICLR 2026_

### Official Review · Reviewer_5a28 · 2025-10-29

**Soundness:** 3
**Presentation:** 3
**Contribution:** 2
**Rating:** 4
**Confidence:** 3

**Summary:**

This paper focuses on testing tool documentation, which is essential for the reliable performance of LLM-based agents. Standard software testing approaches fail to identify documentation-related errors that cause agent tool misuse. To address this gap, the authors propose ToolFuzz, an automated framework for testing tool documentation and uncovering two major types of failures: 1. Tool Runtime Errors and 2. Correctness Failures. ToolFuzz uses two techniques to uncover  errors: (1) by generating queries that lead to tool runtime errors, achieved by combining fuzzing techniques with LLM-based query generation, and (2) by generating queries that result in incorrect agent responses, using synonymous prompt generation and a series of cascading consistency and correctness checks at various stages of the agent’s processing.
The authors also introduce a new benchmark that empirically demonstrates the importance of correct tool documentation, showing that improving documentation based on ToolFuzz findings enhances agent performance on real-world tasks.

**Strengths:**

- The paper is well-organized and clearly written.

- It explores a novel and underexplored problem — testing the documentation of tools rather than just their invocation or usage. Most prior works (I've read) focus on improving tool-use accuracy, while this work identifies the correctness and completeness of documentation as an equally critical factor for reliable agent behavior.

- The paper clearly defines two categories of errors: Tool Runtime Errors and Correctness Failure, and provides intuitive illustrations in Figure 1 that help clarify these distinctions.

- The introduction of a benchmark for real-world agent tasks demonstrates both the practical significance and applicability of the proposed framework. It shows that improving documentation correctness leads to tangible improvements in agent performance.

**Weaknesses:**

- Beyond the two defined error types: Tool Runtime Errors and Correctness Failure, there may exist other classes of documentation issues not covered by ToolFuzz — for instance, semantic ambiguity. The paper could discuss whether and how these might be incorporated in future extensions.

- Both detection pipelines heavily rely on LLMs — for generating prompts, interpreting tool outputs, and serving as oracles. Since results might vary across different LLMs. The paper could strengthen this by introducing rule-based validation components or cross-model agreement metrics to mitigate bias and improve robustness.

- The dependence on the LLM oracle for correctness judgment raises concerns about subjectivity and reproducibility. While effective, this oracle might yield inconsistent results under different LLMs or prompts.

**Questions:**

- As mentioned above, regarding the two main error categories — runtime and correctness failures — do they fully cover all types of tool documentation problems? If not, how might ToolFuzz be extended to capture additional documentation flaws (e.g., ambiguous or incomplete descriptions)?

- Since correctness checking relies on an LLM oracle, have the authors considered more robust alternatives, such as confidence scoring, or rule-based verification? How consistent are the results across different underlying LLMs?

---

> ### Author Response · Authors · 2025-12-03
>
> We thank the reviewer for their comprehensive review and for validating the importance of testing tool documentation alongside our new benchmarks. We address the specific questions regarding documentation error coverage and the robustness of the LLM Oracle below.
>
> **Q1: As mentioned above, regarding the two main error categories — runtime and correctness failures — do they fully cover all types of tool documentation problems? If not, how might ToolFuzz be extended to capture additional documentation flaws (e.g., ambiguous or incomplete descriptions)?**
>
> We draw our error categorization from standard software testing principles. Agentic systems are effectively non-deterministic software; thus, failures generally fall into two buckets: the program crashes (Runtime Failure) or the program runs but produces the wrong result (Correctness Failure).
>
> Documentation issues like ambiguity, under-specification, or over-specification are root causes that manifest as these failures. For example, an ambiguous description often leads to a Correctness Failure where the agent uses the tool inconsistently across synonymous queries. Therefore, while we categorize by failure type, ToolFuzz effectively detects these underlying documentation flaws. Future extensions could explicitly categorize the type of documentation flaw (e.g., "Ambiguity detected") by analyzing the variance in the consistency checks.
>
> **Q2: Since correctness checking relies on an LLM oracle, have the authors considered more robust alternatives, such as confidence scoring, or rule-based verification? How consistent are the results across different underlying LLMs? What about cross-model agreement metrics?**
>
> Yes, we agree that relying solely on an LLM Oracle can introduce subjectivity. However, ToolFuzz is explicitly designed to mitigate this by prioritizing deterministic verification and employing consensus mechanisms before the Oracle is even invoked.
>
> ToolFuzz does not rely on the Oracle as the primary filter. Instead, we prioritize deterministic Consistency Checks (Figure 1.2c). We algorithmically verify that synonymous user queries ($p_1, \dots, p_n$) map to identical tool inputs ($I$) and tool outputs ($O$) via exact matching. This provides an objective, rule-based baseline for error detection that is entirely independent of LLM subjectivity.
>
> We introduce the LLM Oracle only as a secondary "plausibility" filter to handle non-deterministic outputs. Our ablation study (Table 1) demonstrates that this cascading combination is critical: relying solely on the LLM ($\text{TF}_\text{LLM}$) yields a high False Discovery Rate (FDR) of up to 47% (on Composio). However, by cascading the objective consistency checks with the Oracle ($\text{TF}$), we drastically reduce the FDR (e.g., down to 15% on LangChain). This confirms that ToolFuzz uses the Oracle effectively to filter false positives rather than introduce them.
>
> To further address model bias, we employ majority voting when generating expected answers. Additionally, our evaluations across diverse models (GPT-4o, Claude 3.5 Haiku, and Gemini 2.0 Flash) in Appendix H.1 show that ToolFuzz consistently identifies a significant volume of unique errors regardless of the underlying LLM. This demonstrates that the framework is robust and generalizes across models.
>
> **Q3: There may exist other classes of documentation issues not covered by ToolFuzz — for instance, semantic ambiguity. The paper could discuss whether and how these might be incorporated in future extensions.**
>
> Yes. We argue that many aspects leading to semantic ambiguity are covered by the under-specification documentation error types in our paper. Ambiguity comes from the fact that the definition is not precise enough (also see Section 3, paragraph 2).
>
> We cover the underspecification by finding parameters which are not in the proper format for a certain tool i.e. asking what is the weather in SF: this is ambiguous as SF might refer to San Francisco, California or Sofia, Bulgaria and this should be rejected from LLM if the weather tool has a proper specification.

---

### Official Review · Reviewer_uiu4 · 2025-10-31

**Soundness:** 3
**Presentation:** 2
**Contribution:** 3
**Rating:** 4
**Confidence:** 3

**Summary:**

This paper tackles an important and increasingly relevant problem: automated testing of tool-usage reliability in LLM agents. The proposed framework, ToolFuzz, innovatively combines fuzzing with LLM-driven natural language variation to identify runtime and correctness failures in a variety of agent tools. The evaluation shows that many real-world tools do indeed suffer from documentation and behavior inconsistencies, and the ToolFuzz method outperforms the graybox and whitebox baseline methods.

**Strengths:**

1. Addresses a real reliability bottleneck in LLM agents and emerging tool ecosystems, which is well motivated and important.

2. The proposed framework is practical and able to uncover real errors according to the evaluation.

3. Clear problem formulation with a well-engineered pipeline.

**Weaknesses:**

1. Evaluation scope restricted to two tool libraries; generalizability unclear.

2. Heavy reliance on LLM Oracle for consistency check. This may lead to subjective correctness judgments.

3. Writing can be improved. It is a little difficult to capture the novel contribution of the framework when reading Sec. 3 and 4.

**Questions:**

The experiments are mainly conducted with frontier LLMs like GPT and Claude. Have you tried with smaller open-source LLMs?

---

> ### Author Response · Authors · 2025-12-03
>
> We thank the reviewer for their positive assessment and for highlighting the practicality and importance of our framework. We will improve the manuscript's clarity based on the reviewers suggestions. Next, we address all questions raised.
>
> **Q1: The experiments are mainly conducted with frontier LLMs like GPT and Claude. Have you tried with smaller open-source LLMs?**
>
> We primarily utilized frontier models (GPT-4o, Claude 3.5 Sonnet, Gemini 1.5 Pro) because agentic tasks (specifically the reasoning to infer tool usage from documentation) require high capability. Smaller open-source models often struggle with strict instruction following in agentic contexts, leading to high noise in the evaluation.
>
> However, we did evaluate ToolFuzz using GPT-4o-mini, a smaller and more cost-efficient model. As shown in Appendix H.1 (Table 5), ToolFuzz remains effective even with this smaller model, finding 167 unique errors. We anticipate that smaller open-weights models (like Llama 3 8B) could be used, but performance may degrade regarding the precision of the generated "attack" prompts.
>
> **Q2: The evaluation scope is restricted to two tool libraries. The generalizability is unclear.**
>
> We selected these 139 tools specifically because they are open-source, free to use, and do not require authentication or paid API keys. This ensures strict reproducibility of our results, which is often a challenge with benchmarks relying on volatile or paid web APIs (like ToolBench). Expanding the toolset often comes at the cost of reproducibility (due to API changes or paywalls) and accessibility for the research community. However, we believe 139 tools covering diverse domains (LangChain and Composio) provide a sufficient statistical significance and coverage to demonstrate the efficacy of ToolFuzz.
>
> **Q3: Heavy reliance on LLM Oracle for consistency check. This may lead to subjective correctness judgments.**
>
> Yes, we agree that relying solely on an LLM Oracle can introduce subjectivity. However, ToolFuzz is explicitly designed to mitigate this by prioritizing deterministic verification and employing consensus mechanisms before the Oracle is even invoked.
>
> ToolFuzz does not rely on the Oracle as the primary filter. Instead, we prioritize deterministic Consistency Checks (Figure 1.2c). We algorithmically verify that synonymous user queries ($p_1, \dots, p_n$) map to identical tool inputs ($I$) and tool outputs ($O$). This provides an objective, rule-based baseline for error detection that is entirely independent of LLM subjectivity.
>
> We introduce the LLM Oracle only as a secondary "plausibility" filter to handle non-deterministic outputs. Our ablation study (Table 1) demonstrates that this cascading combination is critical: relying solely on the LLM ($\text{TF}_\text{LLM}$) yields a high False Discovery Rate (FDR) of up to 47% (on Composio). However, by cascading the objective consistency checks with the Oracle ($\text{TF}$), we drastically reduce the FDR (e.g., down to 15% on LangChain). This confirms that ToolFuzz uses the Oracle effectively to filter false positives rather than introduce them.
>
> To further address model bias, we employ majority voting when generating expected answers. Additionally, our evaluations across diverse models (GPT-4o, Claude 3.5 Haiku, and Gemini 2.0 Flash) in Appendix H.1 show that ToolFuzz consistently identifies a significant volume of unique errors regardless of the underlying LLM. This demonstrates that the framework is robust and generalizes across models.

---

### Official Review · Reviewer_jm8C · 2025-11-01

**Soundness:** 2
**Presentation:** 3
**Contribution:** 3
**Rating:** 6
**Confidence:** 3

**Summary:**

The paper introduces TOOLFUZZ, an end-to-end agent-centric framework for detecting and fixing tool errors in LLM agents. It uses LLM reasoning with fuzzing to expose runtime and semantic errors. The authors also introduce a new benchmark suite focusing on precise tool invocation for file management and GitHub tasks. TOOLFUZZ is evaluated on LangChain and Composio tools, showing significant improvements in accuracy and error detection.

**Strengths:**

1. The study addresses the underexplored problem of errors in tool documentation.
2. The method implements a structured, end-to-end workflow combining fuzzing, consistency checks, and LLM-based evaluation, ensuring systematic assessment of agent-tool interactions.
3. The experimental evaluation is thorough and well-designed.

**Weaknesses:**

1. The effectiveness of TOOLFUZZ is highly contingent on the reasoning and generative capabilities of the underlying LLM, making it sensitive to model quality.
2. The methodological novelty is limited, as TOOLFUZZ primarily integrates existing techniques without introducing fundamentally new algorithms.
3. The coverage of TOOLFUZZ is constrained, as it may fail to detect errors in tools with complex, context-dependent, or multi-modal inputs and outputs.

**Questions:**

1. Is the selection in the Automatic Documentation Fixing process manually chosen?
2. Can you explain the detailed settings of the variants of TOOLFUZZ?

---

> ### Author Response · Authors · 2025-12-03
>
> We thank the reviewer for their detailed review and for recognizing the systematic nature of our end-to-end workflow. We now clarify the novelty of our pipeline and the robustness of our framework below and address all other points raised by the reviewer.
>
> **Q1: Is the selection in the Automatic Documentation Fixing process manually chosen?**
>
> No, the selection process is fully automated. As detailed in Appendix F, we utilize a split-data approach: For a given tool, ToolFuzz-AutoFix generates 10 candidate documentation fixes. We then evaluate these candidates on a held-out validation set of tasks. The candidate that achieves the highest performance on the validation set is automatically selected. We then report the performance of this selected documentation on the separate test set. There is no manual intervention in selecting the best fix.
>
> We again want to stress that the contribution of our work is primarily the error detection and not the automatic fixing thereof. Rather, ToolFuzz-AutoFix serves as a validation mechanism. Its purpose is to demonstrate that the errors ToolFuzz detects are not merely theoretical noise, but are actionable and useful for improving agent reliability.
>
> **Q2: Can you explain the detailed settings of the variants of ToolFuzz?**
>
> We evaluate different configurations of ToolFuzz to analyze the impact of its components, as shown in Table 1.
> - $\text{TF}$: The complete ToolFuzz pipeline, utilizing both consistency checks and the LLM Oracle.
> - $\text{TF}_\text{CC}$: A variant using only Consistency Checks (checking if synonymous prompts lead to identical tool inputs $I$ and outputs $O$). This variant is cheaper but may miss errors where the agent is consistently wrong.
> - $\text{TF}_\text{LLM}$: A variant using only the LLM Oracle to judge the correctness of the agent's response against an expected answer. This variant typically has a higher False Discovery Rate (FDR) on its own.
>
> The ablation study in Section 5.2 demonstrates that combining these methods (the full TF) achieves the best balance between identifying unique errors and minimizing the False Discovery Rate.
>
> **Q3: The effectiveness of ToolFuzz is highly contingent on the reasoning and generative capabilities of the underlying LLM, making it sensitive to model quality.**
>
> We agree that ToolFuzz leverages the reasoning capabilities of the underlying LLM. However, our empirical results demonstrate that the threshold for "sufficient capability" is well met by current, widely available models, including smaller, cost-efficient ones.
>
> As detailed in Appendix H.1 (Table 5), we evaluated ToolFuzz across a diverse set of models, including GPT-4o, GPT-4o-mini, Claude 3.5 Haiku, and Gemini 2.0 Flash. We found that ToolFuzz consistently identifies a high volume of unique errors across all tested models. Notably, the smaller GPT-4o-mini actually identified slightly more unique runtime errors (167) than the larger GPT-4o (143) in some configurations. This indicates that ToolFuzz is not brittle; it does not require the absolute frontier model to be effective.
>
> While ToolFuzz is already effective with current models, the rapid and continuous improvement of LLM capabilities will also benefit our approach. As models become cheaper and more capable, ToolFuzz will naturally become even more efficient at detecting subtle errors without requiring architectural changes to the fuzzing framework itself.

---

> ### Author Response · Authors · 2025-12-03
>
> **Q4: The methodological novelty is limited, as ToolFuzz primarily integrates existing techniques without introducing fundamentally new algorithms.**
>
> ToolFuzz introduces significant novelty in problem formulation and system design, specifically through the novel adaptation of metamorphic testing to the domain of GenAI agents. We address this in three key points:
>
> 1. Novel Application of Metamorphic Testing: While the component techniques (fuzzing, LLM generation) exist, their integration into our Correctness Failure Detection pipeline (Figure 1.2) represents a fundamental methodological innovation. We solve the "missing oracle" problem for unknown tools (where ground truth is rarely available) by introducing an invariance-based testing approach. By generating sets of synonymous prompt templates ($P$) and enforcing that they map to invariant tool inputs ($I$) and outputs ($O$), we created a novel method to detect semantic documentation errors automatically. This is a non-trivial adaptation of metamorphic testing principles to the semantic, non-deterministic space of LLM agents.
> 2. Novel Problem Formulation: ToolFuzz is the first automated method specifically formulated to test agent tool documentation. Prior work has focused on testing the LLM's reasoning capabilities or the tool’s source code in isolation. Our contribution lies in defining and rigorously testing the interaction contract between the two. We formalized specific failure modes in this context: under-specification, over-specification, and ill-specification and then built a targeted framework to detect them. Previous API testing methods fail to capture because they ignore the natural language "interface" used by the agent.
> 3. Efficacy via Systemic Integration: We view the seamless integration of these techniques as a feature that enhances robustness rather than a limitation. The primary metric for a testing framework is its ability to find bugs. As demonstrated in Table 1, our specific integration allows ToolFuzz to outperform baselines by identifying 50% more unique errors while reducing the False Discovery Rate (FDR) by 4.5x. This demonstrates that our specific algorithmic choices (e.g., the cascading checks in Figure 1) provide a substantial advancement over the state-of-the-art in reliable agent construction.
>
> **Q5: The coverage of ToolFuzz is constrained, as it may fail to detect errors in tools with complex, context-dependent, or multi-modal inputs and outputs.**
>
> We want to clarify that ToolFuzz is explicitly designed to handle context-dependent and complex structured inputs better than traditional methods.
>
> Contrary to the reviewer’s concern that ToolFuzz might fail here, our hybrid architecture (Section 4.1) was built specifically to address this challenge. We employ Taint Analysis to extract strict syntax priors (e.g., JSON formats, specific string structures). This ensures that we generate complex, structurally valid inputs that pass basic validation but stress-test the deeper logic. We do not rely on "dumb" fuzzing. We utilize an LLM for prompt generation specifically to wrap fuzz-generated arguments into semantically and contextually appropriate natural language queries. Furthermore, we leverage semantic priors (dictionaries, LMs) during the fuzzing stage itself. This ensures the inputs are not just random noise, but context-aware edge cases.
>
> Regarding multi-modal inputs: The current implementation focuses on text and structured data (JSON/Code), as these represent the vast majority of current agentic tool interfaces (APIs, SQL, Function Calling). However, the methodological contribution like the Correctness Pipeline (Figure 1.2) is modality-agnostic. The principle of Metamorphic Testing (verifying that synonymous inputs yield invariant tool invocations) holds true whether the input is text, an image, or audio. Extending the generator to produce multi-modal inputs is a valid engineering task for future work, but the core logic remains applicable.
>
> We focus with ToolFuzz on the atomic contract (tool description) of the agent-tool interaction. Robustly testing this is a prerequisite for reliable complex interactions. As noted in our Conclusion, ToolFuzz solves the fundamental problem of individual tool reliability; without this foundation, testing complex multi-modal chains is prone to noise and attribution errors.

---

### Official Review · Reviewer_pAdL · 2025-11-02

**Soundness:** 3
**Presentation:** 3
**Contribution:** 2
**Rating:** 4
**Confidence:** 4

**Summary:**

To address LLM Agent unreliability from flawed tool documentation (under-, over-, or ill-specified), this paper proposes "TOOLFUZZ," the first automated agent tool testing method. It combines fuzzing with LLM-driven query generation to detect runtime failures and correctness failures, the latter via an invariance-based approach (synonymous prompts and cascading consistency checks). Evaluated on 139 LangChain and Composio tools, TOOLFUZZ found 50% more unique errors than baselines while reducing the False Discovery Rate (FDR) by 4.5x. Furthermore, auto-fixing documentation using TOOLFUZZ's findings significantly improved agent accuracy on two novel benchmarks , raising GitHub task accuracy from 22.9% to 35.4%.

**Strengths:**

1.	Based on fuzzing and LLM-driven prompt generation, this paper proposes an automated testing method for LLM Agent tools, capable of systematically discovering both runtime failures and correctness errors in tools. By identifying these defects and using them to automatically optimize tool documentation, the method significantly boosts the agent's accuracy in handling tool calls; for example, accuracy on GitHub tasks increased from a baseline of 22.9% to 35.4%, demonstrating its effectiveness in enhancing agent reliability.
2.	Furthermore, this paper introduces two novel benchmarks—File Management and GitHub tasks—specifically designed to evaluate tool-call accuracy, filling a gap in existing agent benchmarks.

**Weaknesses:**

1.	TOOLFUZZ is fundamentally limited to testing single-tool accuracy, failing to capture critical agent failures that arise from complex multi-step planning, intermediate multi-tool calls, or inter-API dependencies. Furthermore, the reliance on artificially synthesized "malicious queries" introduces an input bias, which undermines the generalizability of the static document fixes.
2.	Testing only 139 tools is insufficient to represent the vast and rapidly evolving LLM Agent API ecosystem, limiting the applicability of the findings. Crucially, even after document optimization, the final absolute accuracy remains below 40% (e.g., 35.4% on GitHub tasks). This low ceiling suggests the method primarily addresses basic syntactical and obvious semantic flaws, offering only a marginal boost to overall tool-call precision.
3.	The optimization is static and pre-deployment, lacking a mechanism for continuous, dynamic adaptation based on experience (e.g., DRAFT). This inability to learn from real-world long-tail errors, combined with a reliance on Prompt Engineering over deeper technical methods like SFT or RL, results in a shallower overall technical contribution.

**Questions:**

1.	How do you ensure your method's generated tests cover the majority of potential LLM errors, since many errors originate from multi-step planning, etc.?
2.	Your method lacks a dynamic optimization mechanism. How can the tool documentation be continuously updated and refined based on long-tail errors or experiences encountered during actual interaction?
3.	Despite testing 139 tools, this number is insufficient. Can you expand the toolset?

---

> ### Author Response · Authors · 2025-12-03
>
> We thank the reviewer for their constructive feedback and for recognizing ToolFuzz as an innovative approach that effectively enhances agent reliability. We appreciate the points raised regarding the scope of single-tool testing and dynamic optimization, which we address in detail below.
>
> **Q1: How do you ensure your method's generated tests cover the majority of potential LLM errors, since many errors originate from multi-step planning, etc.?**
>
>
> We acknowledge that agent failures can stem from complex multi-step planning. However, ToolFuzz focuses on the atomic unit of agent interaction: the tool invocation itself. If an individual tool is documented incorrectly (leading to runtime crashes or semantic misunderstandings), the planner cannot possibly succeed, regardless of its reasoning capabilities.
>
> While ToolFuzz does not explicitly test the long-horizon planning logic, it ensures the necessary prerequisites for successful planning are met. By fixing the "contracts" (documentation) between the LLM and the tools, we remove the root cause of many planning failures. As noted in our conclusion, extending ToolFuzz to test multi-tool interactions and chained dependencies is a promising direction but needs to be left to future work.
>
> **Q2: Your method lacks a dynamic optimization mechanism. How can the tool documentation be continuously updated and refined based on long-tail errors or experiences encountered during actual interaction?**
>
> We emphasize that ToolFuzz is fundamentally designed as a fuzzer, not a fixer (like DRAFT or EasyTool). We address the distinction in two key ways:
>
> The core goal of our work is ToolFuzz, which aims to maximize error discovery by proactively exploring a wide variety of potential inputs (fuzzing). The documentation fixing pipeline presented in the paper (ToolFuzz-AutoFix) is not the primary contribution. Rather, it serves as a validation mechanism: Its purpose is to demonstrate that the errors ToolFuzz detects are not merely theoretical noise, but are actionable and useful for improving agent reliability.
>
> Dynamic optimization mechanisms often react to a single failure instance encountered during interaction. This approach risks overfitting the documentation to the specific long-tail error of the moment, potentially degrading general performance or breaking other functionalities. In contrast, ToolFuzz aims for broad exploration to ensure robustness before deployment. Our experiments support this: when applying dynamic fixing (DRAFT) on top of our fixes, performance actually degraded in some cases (Table 2), suggesting that overfitting to specific instances can be detrimental.
>
> ToolFuzz is designed as a pre-deployment static testing framework meant to identify vulnerabilities early, similar to classical software fuzzers like AFL. While dynamic optimization is reactive (fixing errors after they occur in production), ToolFuzz is proactive. However, the feedback loop ToolFuzz-Autofix allows for can certainly be integrated into a pipeline to continuously regress-test documentation as new edge cases are discovered.
>
> **Q3: Despite testing 139 tools, this number is insufficient. Can you expand the toolset?**
>
> We selected these 139 tools specifically because they are open-source, free to use, and do not require authentication or paid API keys. This ensures strict reproducibility of our results, which is often a challenge with benchmarks relying on volatile or paid web APIs (like ToolBench). Expanding the toolset often comes at the cost of reproducibility (due to API changes or paywalls) and accessibility for the research community. However, we believe 139 tools covering diverse domains (LangChain and Composio) provide a sufficient statistical significance and coverage to demonstrate the efficacy of ToolFuzz.

---

### Meta-Review · Area_Chair_m1i8 · 2025-12-14

**Summary:**

The problem of proactively testing LLM agent tool documentation is both important and underexplored. ToolFuzz offers a compelling contribution through its practical impact, solid engineering, and the introduction of novel, well-motivated benchmarks. However, to further strengthen the work, it would be valuable to expand the scope of tool coverage and more clearly demonstrate the scalability of the framework across a wider range of tool ecosystems. Additionally, while the authors address concerns around subjectivity by incorporating cascading consistency checks, a more rigorous quantitative evaluation of the framework’s reliance on the LLM Oracle---particularly in terms of alignment with ground truth, consistency across prompts, and robustness across different models---would improve confidence in its generalizability and reliability.

**Reviewer Concerns:**

The rebuttal effectively addressed several reviewer concerns. Reviewer pAdL’s concern regarding the lack of support for multi-step, multi-tool scenarios and the use of static optimization was met with a clear explanation of the framework’s focus on atomic tool reliability as a foundational step. Reviewer 5a28’s question about whether runtime and correctness failures sufficiently cover documentation issues was reasonably addressed by framing these failures within standard software testing principles.

However, some concerns remain partially unresolved. Reviewers’ concerns about the limited scope of tool coverage and the generalizability of the framework across a broader agent tool ecosystem were acknowledged but not fully resolved; expanding beyond the current set of 139 tools would help substantiate claims of scalability. Additionally, while the authors provided qualitative justification and empirical results across diverse models, a more rigorous quantitative evaluation of the framework’s reliance on the LLM Oracle, particularly regarding alignment with ground truth and consistency across prompts, would further strengthen confidence in the robustness and general applicability of ToolFuzz.

**Reviewer Scores:**

Reviewer pAdL’s concerns, particularly regarding the handling of multi-step, multi-tool workflows and the focus on static documentation, were largely addressed in the rebuttal through clarifications about the framework’s current scope and future extensibility. As a result, they would likely increase their score from 4 to 6.

Reviewer jm8C expressed a generally positive view of the work and had concerns primarily around subjectivity and automation, both of which were effectively addressed in the rebuttal. They would likely maintain their positive score, possibly reaffirming a 6.

Reviewers uiu4 and 5a28 raised concerns about the limited tool coverage, generalizability, and reliance on LLMs. While the authors provided reasonable justifications to address these points, these responses may not have been sufficient to fully alleviate their concerns. As such, both reviewers would likely maintain their original scores.

---

### Decision · Program_Chairs · 2026-01-26

Reject